# Feature-Proxy Transformer for Few-Shot Segmentation

**Jian-Wei Zhang**[1]*,   **Yifan Sun**[2],    **Yi Yang**[3],    **Wei Chen**[1]†

[1] State Key Lab of CAD&CG, Zhejiang University, Hangzhou, China
[2] Baidu Research
[3] CCAI, College of Computer Science and Technology, Zhejiang University
{zjw.cs,yangyics,chenvis}@zju.edu.cn, sunyf15@tsinghua.org.cn

## Abstract

Few-shot segmentation (FSS) aims at performing semantic segmentation on novel classes given a few annotated support samples. With a rethink of recent advances, we find that the current FSS framework has deviated far from the supervised segmentation framework: Given the deep features, FSS methods typically use an intricate decoder to perform sophisticated pixel-wise matching, while the supervised segmentation methods use a simple linear classification head. Due to the intricacy of the decoder and its matching pipeline, it is not easy to follow such an FSS framework. This paper revives the straightforward framework of "feature extractor + linear classification head" and proposes a novel Feature-Proxy Transformer (FPTrans) method, in which the "proxy" is the vector representing a semantic class in the linear classification head. FPTrans has two keypoints for learning discriminative features and representative proxies: 1) To better utilize the limited support samples, the feature extractor makes the query interact with the support features from bottom to top layers using a novel prompting strategy. 2) FP-Trans uses multiple local background proxies (instead of a single one) because the background is not homogeneous and may contain some novel foreground regions. These two keypoints are easily integrated into the vision transformer backbone with the prompting mechanism in the transformer. Given the learned features and proxies, FPTrans directly compares their cosine similarity for segmentation. Although the framework is straightforward, we show that FPTrans achieves competitive FSS accuracy on par with state-of-the-art decoder-based methods. [1]

## 1   Introduction

Few-shot learning is of significant value for semantic segmentation. It is because the semantic segmentation task requires pixel-wise annotation, which is notoriously cumbersome and expensive [10, 30, 28]. Therefore, learning from very few samples for semantic segmentation has attracted significant research interest, yielding a popular topic, *i.e.*, few-shot semantic segmentation (FSS). Formally, FSS aims at performing semantic segmentation on novel classes given only a few (*e.g.*, one or five) densely-annotated samples (called *support* images) [5].

With a rethink of recent advances in FSS, we find current FSS methods usually require an intricate decoder, deviating far from the plain supervised segmentation framework. More concretely, state-of-the-art FSS methods adopt the "feature extractor + (intricate) decoder" framework (Fig. 1 (a), (b) and

---

*Work done during an internship at Baidu Research.

†Corresponding author.

[1]Code is available at https://github.com/Jarvis73/FPTrans.

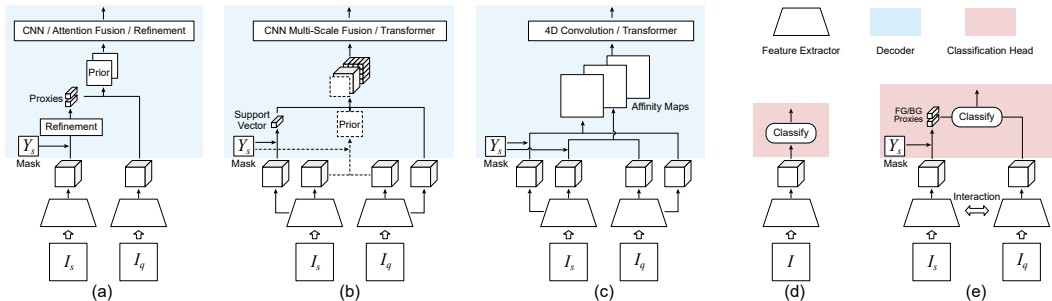

Figure 1: Comparison between the "feature extractor + intricate decoder" (a, b and c) and the plain "feature extractor + linear classification head" framework (d and e). **(a)** The decoder refines the prior maps (raw segmentation) for final prediction (*e.g.*, FWB [38], CWT [36], SCL [61]). **(b)** The decoder concatenates the support prototype and the query features and then further feeds them into the CNN or transformer (*e.g.*, PFENet [46], ASGNet [27], CyCTR [64]). **(c)** The decoder conducts pixel-to-pixel matching from query to support and then applies 4D convolution or transformer to discover the patterns within the matching score maps (*e.g.*, HSNet [37], VAT [19]). **(d)** The plain framework of "feature extractor + linear classification head" in supervised segmentation. **(e)** The proposed FPTrans revives the plain framework and makes only one necessary modification (*i.e.*, the proxies are extracted from the support images on the fly).

(c)), while the supervised segmentation methods usually adopt the "feature extractor + (simple) linear classification head" framework (Fig. 1 (d)). In FSS frameworks, the decoders perform sophisticated matching and can be summarized into three types (Fig. 1 (a), (b), and (c)), as detailed in the related works in Section 2.1. Arguably, the intricacy of the decoder and its sophisticated matching pipeline makes the FSS framework hard to follow. Under this background, we think it is valuable to explore a relatively straightforward FSS framework.

This paper revives the plain framework of "feature extractor + linear classification head" for FSS and proposes a novel Feature-Proxy Transformer (FPTrans) method. The term "proxy" denotes the vector representing a foreground class or the background in the linear classification head. We note that in Fig. 1 (d), given the extracted feature maps, the supervised segmentation methods simply feed them into a linear classification head to perform pixel-wise prediction. FPTrans adapts this straightforward framework to the FSS task with only one modification: instead of fixing the already-learned proxies in the classification head (Fig. 1 (d)), FPTrans uses support feature maps and the support mask to generate the proxies on the fly. This modification is necessary for recognizing novel classes and is consistent with some earlier FSS methods [41, 50].

To tackle two FSS challenges (*i.e.*, generalization to novel classes and very few support samples) under this simple framework, FPTrans has two keypoints for learning discriminative features and representative proxies, respectively. **1)** To better utilize the limited support samples, the feature extractor makes the query interact with the support features from the bottom to top layers. Consequently, the support sample provides extra information/clues for extracting the query features and is thus beneficial. **2)** To promote generalization to novel classes, FPTrans uses multiple local background proxies instead of a single global proxy for representing the background. This design is important because, during base training, the background is not homogeneous and may contain some novel foreground classes. Consequently, it avoids confusing novel classes with the background of the base data and thus benefits generalization to novel classes.

We implement the above two keypoints with a novel prompting strategy. While using the prompt to condition a transformer for different tasks [29, 23, 51] or different domains [13] is a common practice, our prompting strategy is significantly different and has two novel functions, *i.e.*, 1) prompting for different (foreground or background) proxies and, 2) acting as the intermediate for query-support interaction. Specifically, FPTrans simultaneously prepends multiple prompts at the input layer, with one prompt for the foreground and the other prompts for the background. These prompts are fed into the transformer and finally become the foreground proxy and local background proxies, respectively. During their flow from the input layer to the output layer, the hidden states of prompts in all the hidden layers are shared by the query and support for cross-attention (query-prompt attention and support-prompt attention). Therefore, it significantly reduces the interaction complexity from

$O(N^2)$ to $O(N)$ ($N$ is the number of pixels on the feature maps). Since the prompting and attention mechanism are critical for these two keypoints, we use the transformer backbone as a natural choice.

We conduct extensive experiments and show that FPTrans achieves accuracy on par with the decoder-based FSS methods. For example, on PASCAL-$5^i$ [5] with one support sample, FPTrans achieves 68.81% mIoU, setting a new state of the art. With its simple framework and competitive accuracy, we hope FPTrans can serve as a strong baseline for FSS.

To sum up, our main contributions are summarized as follows: (i) We revive the plain "feature extractor + linear classification head" framework for FSS. We hope the correspondingly-proposed method FPTrans can serve as a simple and strong FSS baseline. (ii) We integrate two keypoints into FPTrans, *i.e.*, learning discriminative features through query-support interaction and learning representative local background proxies. These two keypoints rely on a novel prompting strategy of the transformer backbone and correspondingly tackle two FSS challenges, *i.e.*, very few support samples and generalization to novel classes. (iii) We conduct extensive experiments to validate the effectiveness of FPTrans. Experimental results show that FPTrans with a plain framework achieves competitive accuracy, compared with state-of-the-art FSS methods with intricate decoders.

## 2 Related Work

### 2.1 Recent Progress on Few-Shot Segmentation

Early FSS methods directly generate class-specific classifier weights and biases [41, 40, 45] or apply the prototype learning [44, 8, 50, 42]. The recent state-of-the-art methods shift to the decoder-based framework and can be categorized into three types according to the decoder structure. **1)** Some recent methods[38, 32, 36, 61] (Fig. 1(a)) generate prior maps based on query features and support features. They further use CNN or transformer to refine the prior maps (which may be viewed as raw segmentation) into the final segmentation output. **2)** Some methods [66, 63, 46, 27, 64, 54, 55, 56, 62] (Fig. 1(b)) concatenate the support prototype vector and the query features and then feed the concatenated feature maps into a subsequential CNN or transformer for prediction. **3)** Some methods focus on exploring fine-grained knowledge and calculating the pixel-to-pixel matching scores between support and query features to derive affinity maps [21, 49, 58, 11, 64, 59, 36] (Fig. 1(c)). Some further apply 4D convolution or transformer [37, 19] to discover the latent patterns within the affinity maps. Besides, latent information is also investigated to enhance the model [39, 67, 25, 57, 52, 34].

In contrast, this paper abandons the intricate decoder and revives the plain framework of "feature extractor + linear classification head". We show that this simple framework can also achieve promising FSS results. We note that a recent work [2] also uses the plain framework during training. However, they rely on transductive inference during testing to mitigate the gap between the inconsistent training and testing schemes. Compared with [2], the proposed FPTrans maintains its simplicity for testing and achieves superior FSS accuracy.

### 2.2 Backbone for Few-Shot Segmentation

Previous FSS methods usually adopt CNNs (*e.g.*, VGG [43], ResNet [17]) as the backbone (*i.e.*, feature extractor) and typically fix the pretrained backbone parameters [63, 46, 64, 61, 52, 36, 27, 37, 19]. However, fixing the backbone is prone to a side-effect, *i.e.*, insufficient adaptation to the segmentation training data. Some recent methods finetune the CNN backbone along with FSS training with sophisticated techniques (*e.g.*, model ensemble or transductive inference [26, 2]) to tackle the problem of insufficient adaptation.

In contrast, this paper adopts the vision transformer [9, 18, 53, 35] as the backbone because we rely on the attention mechanism and a novel prompting technique for query-support interaction. Moreover, the proposed FPTrans benefits from fine-tuning the backbone parameters without bells and whistles. We note that using the transformer backbone does NOT necessarily improve FSS, because the ablation studies (in Table 5) show that replacing the CNN backbone with a transformer does NOT bring improvements to the decoder-based FSS methods [46, 64]. Therefore, we attribute the superiority of our method mainly to the two unique keypoints in FPTrans.

# 3 Methods

## 3.1 Problem Formulation

Few-shot segmentation aims at tackling the semantic segmentation problem on novel classes under a low data regime. Specifically, FSS usually provides a training set with categories $\mathcal{C}_{train}$ and a testing set with novel categories $\mathcal{C}_{test}$ ($\mathcal{C}_{train} \cap \mathcal{C}_{test} = \emptyset$). The mainstream setting [63, 64, 46] adopts the episodic training and testing scheme: Each episode corresponds to a single class $c$ ($c \in \mathcal{C}_{train}$ during training and $c \in \mathcal{C}_{test}$ during testing), and provides a query sample $\{I_q, Y_q\}$ and $K$ support samples $\{I_s^{(k)}, Y_s^{(k)}\}_{k=1}^K$ ($I$ is the image and $Y$ is the label). The superscript $k$ will be omitted unless necessary. Within each episode, the model is expected to use $\{I_s, Y_s\}$ and $I_q$ to predict the query label. In this paper, we follow this popular episodic training and testing scheme.

## 3.2 Preliminaries on Vision Transformer and Prompt Learning

The proposed FPTrans uses the vision transformer as its backbone and integrates a novel prompting strategy. Therefore, we first revisit the vision transformer and the prompting mechanism.

**Vision Transformer** [9](ViT) is designed for computer vision tasks based on transformer [48] which is originally designed for sequential data [7, 3, 33]. It is composed of a patch embedding module, a transformer encoder, and an MLP head. Given an RGB image as the input, ViT first reshape it into $N$ patches $\{\mathbf{a}_p \in \mathbb{R}^{3 \times P \times P} | p = 1, 2, \ldots, N\}$ ($P$ is the patch size) and then projects the flattened image patches into $C$-Dimensional embeddings by $\mathbf{x}_p = \text{Embed}(\mathbf{a}_p) \in \mathbb{R}^C$, $p = 1, 2, \ldots, N$. We denote the collection of these embedding tokens as $\mathbf{X}^0 = \{\mathbf{x}_p\}_{p=1}^N \in \mathbb{R}^{N \times C}$. ViT has $L$ stacked transformer blocks, each one of which consists of a Multiheaded Self-Attention module and an MLP module (See supplementary Section A for details). Given $\mathbf{X}^0$ (the collection of embedding tokens), ViT concatenates it with a classification token $\mathbf{x}_{cls}^0 \in \mathbb{R}^C$ and then inputs them into the stacked transformer blocks, which is formulated as:

$$[\mathbf{x}_{cls}^l, \mathbf{X}^l] = B_l([\mathbf{x}_{cls}^{l-1}, \mathbf{X}^{l-1}]), \qquad l = 1, 2, \ldots, L, \tag{1}$$

where $\mathbf{x}_{cls}^l$ and $\mathbf{X}^l$ are the outputs of the $l$-th block $B_l$, and $[\cdot, \cdot]$ is the concatenate operation.

**Prompting** was first introduced in NLP tasks to identify different tasks by inserting a few hint words into input sentences [12, 24, 4, 31]. More generally, prompting techniques can efficiently condition the transformer to different tasks [29, 14, 20, 16, 51] or domains [13] without changing any other parameters of the transformer. To this end, prompting techniques typically prepend some prompt tokens $\mathbf{P}^0$ to the input layer. Correspondingly, Eqn (1) transforms into $[\mathbf{x}_{cls}^l, \mathbf{X}^l, \mathbf{P}^l] = B_l([\mathbf{x}_{cls}^{l-1}, \mathbf{x}^{l-1}, \mathbf{P}^{l-1}])$. It can be seen that changing the prompt simultaneously changes the mapping function of the transformer, even if all the transformer blocks $B_l$ remain unchanged.

**Our prompting strategy** in FPTrans is significantly different from the prior prompting techniques. In contrast to the popular prompting for different tasks or different domains, FPTrans prepends multiple prompts to simultaneously activate multiple different proxies (*i.e.*, the foreground and local background proxies), as well as to facilitate efficient query-support interactions. Moreover, in prior works, the prompts in the hidden layers are dependent on a single input sample. In contrast, in FPTrans, the prompts in the hidden layers are shared by the query and support images through a synchronization procedure. We will illustrate these points in the following section.

## 3.3 Feature-Proxy Transformer

### 3.3.1 Overview

The proposed Feature-Proxy Transformer (FPTrans) consists of three major steps, *i.e.*, 1) prompt generation, 2) feature and proxy extraction, and 3) training the classification head and inference, as illustrated in Figure 2. The prompts are extracted from the support images within a (training or testing) episode and contain a foreground prompt and multiple local background prompts (Section 3.3.2). Afterward, FPTrans concatenates these prompts with the image patch tokens and forwards them into the stacked transformer blocks to extract both the features and proxies (Section 3.3.3). Different from the popular prompting technique, FPTrans shares the hidden states of the prompt tokens for the query and its support images, using a prompt synchronization operation. It facilitates efficient feature

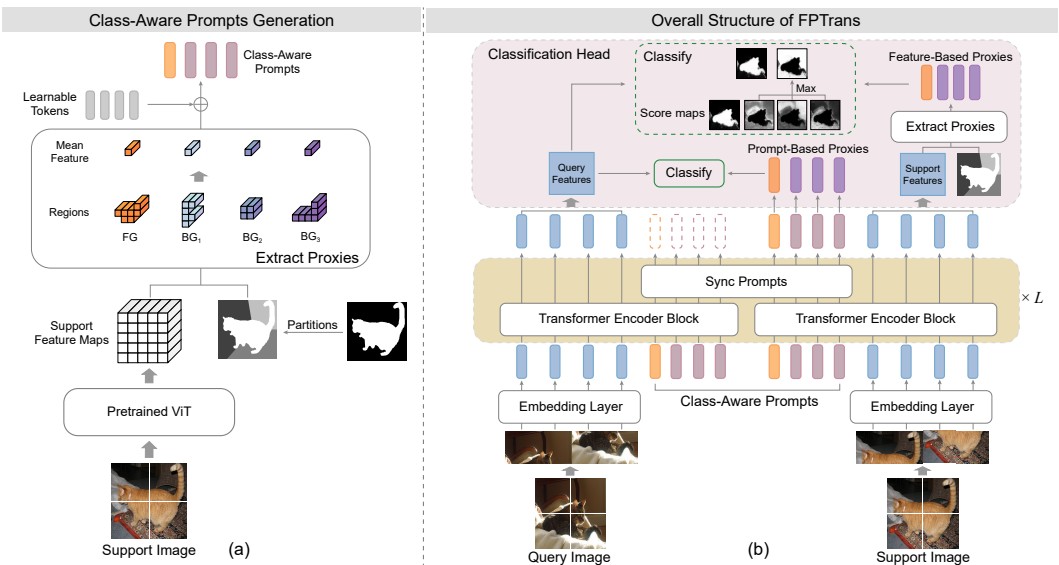

Figure 2: Overview of the proposed Feature-Proxy Transformer (FPTrans). **(a)** Given a support image, we generate a foreground prompt and multiple local background prompts. Each prompt consists of multiple tokens. **(b)** The feature extractor consists of $L$ transformer blocks. It takes patch tokens from the query and support images, as well as the prompts as its input. After every transformer block, FPTrans synchronizes the prompt tokens from the query and support branches to facilitate efficient query-support interactions. The classification head uses two types of proxies (the feature-based and the prompt-based proxies) for training and uses the feature-based proxies for inference.

interactions between support and query features. Finally, given the extracted features and proxies, we elaborate on how to use them for training the classification head and for inference in Section 3.3.4.

### 3.3.2 Prompt Generation

Prompt generation is illustrated in Fig. 2 (a). Within each episode, FPTrans uses the support image(s) to extract class-aware prompts and share them for the support and query images. The prompts are class-aware because we respectively extract prompts from the foreground and background.

To this end, we first use a pretrained plain vision transformer (Eqn. (1)) to extract deep features from the support image and get $\mathbf{F}_s^* \in \mathbb{R}^{C \times H \times W}$, in which $H$ and $W$ are the feature height and width, and the subscript $s$ indicates the "support". Afterward, we use the support mask to crop out the foreground regions and background regions from $\mathbf{F}_s^*$. While the foreground regions are cropped out as a whole, the background regions are cropped and partitioned into multiple ($S$) local regions, because the background is likely not to be homogeneous. To this end, we employ a Voronoi-based method [1] to partition the background into $S$ regions (See supplementary Section B.1 for details).

According to the background partition results, we generate $S$ local masks for the background, *i.e.*, $\{B_n\}_{n=1}^{S}$ with $B_n \in \{0, 1\}^{H \times W}$. A partition example of $S = 3$ is illustrated in Fig. 2(a). Consequently, we calculate the mean feature of the foreground $\mathbf{u}_f^* \in \mathbb{R}^C$ and the local mean feature of background $\mathbf{u}_n^* \in \mathbb{R}^C$ by masked average pooling:

$$\mathbf{u}_f^* = \frac{1}{|\tilde{Y}_s|} \sum_{i=1}^{HW} \mathbf{F}_{s,i}^* \tilde{Y}_{s,i}, \quad \mathbf{u}_n^* = \frac{1}{|B_n|} \sum_{i=1}^{HW} \mathbf{F}_{s,i}^* B_{n,i}, \qquad n = 1, \dots, S, \tag{2}$$

where $\tilde{Y}_s \in \{0, 1\}^{H \times W}$ is the down-sampled foreground mask of the support image.

Based on the mean features, we further expand each $C$-dimensional mean feature vector ($\mathbf{u}_f^* \in \mathbb{R}^C$ and $\mathbf{u}_n^* \in \mathbb{R}^C$) into a corresponding $G \times C$ token and then add it with extra learnable tokens by:

$$\mathbf{p}_f = \mathcal{E}(\mathbf{u}_f^*) + \mathbf{z}_f, \quad \mathbf{p}_n = \mathcal{E}(\mathbf{u}_n^*) + \mathbf{z}_n, \qquad n = 1, \dots, S, \tag{3}$$

where $\mathcal{E}$ is the expansion from $C$-dimensional vector to $G \times C$-dimensional token, and $\mathbf{z}_f, \mathbf{z}_n \in \mathcal{R}^{G \times C}$ are the learnable tokens. Adding these learnable tokens makes prompts gain extra diversity and become more discriminative [60]. See supplementary Section B.2 for more details on this prompt augmentation technique. Consequently, after the expansion and augmentation, we get a foreground prompt token $\mathbf{p}_f \in \mathcal{R}^{G \times C}$ and $S$ local background prompt tokens $\mathbf{p}_n \in \mathcal{R}^{G \times C}(n = 1, 2, \cdots, S)$.

### 3.3.3 Feature and Proxy Extraction

**Feature Extraction.** The process for extracting features is illustrated in Fig. 2(b). Without loss of generality, we take the 1-shot setting as the example for clarity. (See supplementary Section C for $K$-shot settings) Within each episode, a query image $I_q$ and a support image $I_s$ are split into $N$ patches and then flattened as $\{\mathbf{a}_{q,p}\}_{p=1}^N$ and $\{\mathbf{a}_{s,p}\}_{p=1}^N$, respectively. An embedding layer projects these patches into query and support patch tokens, $i.e.$, $\mathbf{X}_q^0 \in \mathbb{R}^{N \times C}$ and $\mathbf{X}_s^0 \in \mathbb{R}^{N \times C}$. We recall that in Section 3.3.2, we already get multiple prompt tokens, $i.e.$, $\mathbf{P}^0 = [\mathbf{p}_f, \mathbf{p}_0, \mathbf{p}_1, \cdots, \mathbf{p}_S]$. The patch tokens from a single image and the prompt tokens are concatenated as the query input and support input, $i.e.$, $[\mathbf{X}_q^0, \mathbf{P}^0]$ and $[\mathbf{X}_s^0, \mathbf{P}^0]$. The query and support inputs are processed by the transformer blocks, which are formulated by:

$$[\mathbf{X}_q^l, \mathbf{P}_q^l] = B_l([\mathbf{X}_q^{l-1}, \mathbf{P}^{l-1}]), \tag{4}$$

$$[\mathbf{X}_s^l, \mathbf{P}_s^l] = B_l([\mathbf{X}_s^{l-1}, \mathbf{P}^{l-1}]), \tag{5}$$

$$\mathbf{P}^l = (\mathbf{P}_q^l + \mathbf{P}_s^l)/2, \tag{6}$$

where $l = 1, 2, \cdots, L$ enumerates all the transformer blocks.

There is a novel and unique **prompt synchronization** (Eqn. (6)) in FPTrans. Specifically, Eqn. (6) averages the query and support prompt tokens after every transformer block. It makes the synced prompt token $\mathbf{P}_l$ absorb information from both the query and support patch tokens. Therefore, in the subsequential $(l + 1)$-th block, the synced prompt token $\mathbf{P}_l$ passes the support information to the query tokens $[\mathbf{X}_q^{l+1}, \mathbf{P}_q^{l+1}]$ and vice versa. In a word, this simple prompt synchronization facilitates efficient interaction between query and support features.

**Proxy Extraction.** As the results of Eqn. (4) to Eqn. (6), the feature extractor outputs support features $\mathbf{X}_s^L$, query features $\mathbf{X}_q^L$, as well as the deep states of prompts $\mathbf{P}^L$. FPTrans uses the support features $\mathbf{X}_s^L$ and the deep states of prompts $\mathbf{P}^L$ to extract two types of proxies, $i.e.$, the feature-based proxies and the prompt-based proxies, respectively.

• *Feature-based proxies*. FPTrans uses the downsampled foreground mask $\tilde{Y}_s$ and the partitioned background masks $B_n$ (which are already provided in Section 3.3.2) to extract the feature-based proxies by: $\mathbf{u}_f = \frac{1}{|\tilde{Y}_s|} \sum_i \mathbf{F}_{s,i} \tilde{Y}_{s,i}$, $\mathbf{u}_n = \frac{1}{|B_n|} \sum_i \mathbf{F}_{s,i} B_{n,i}$ $n = 1, \ldots, S$.

• *Prompt-based proxies*. Moreover, FPTrans uses the deep states of prompts $\mathbf{P}^L$ to get the prompt-based proxies by: $\mathbf{v}_f = \frac{1}{G} \sum_{j=1}^G \mathbf{p}_{f,j}^L$ and $\mathbf{v}_n = \frac{1}{G} \sum_{j=1}^G \mathbf{p}_{n,j}^L$.

These two types of proxies are both used for training FPTrans. In contrast, during testing, FPTrans only uses feature-based proxies.

### 3.3.4 Training and Inference

FPTrans uses two classification losses (one for the feature-based proxies and the other for the prompt-based proxies) and a pairwise loss for training.

**Classification losses.** Since segmentation can be viewed as a pixel-wise classification problem, FPTrans directly compares the similarity of each query feature vector ($\mathbf{F}_{q,i}$) and the proxies for linear classification. We elaborate on the classification loss with the feature-based proxy (and the process with the prompt-based proxies is similar). The predicted probability of $\mathbf{F}_{q,i}$ belonging to the foreground is formulated as:

$$\mathcal{P}(\mathbf{F}_{q,i}) = \frac{\exp(\texttt{sim}(\mathbf{F}_{q,i}, \mathbf{u}_f)/\tau)}{\exp\left(\texttt{sim}(\mathbf{F}_{q,i}, \mathbf{u}_f)/\tau\right) + \max_n\left(\exp\left(\texttt{sim}(\mathbf{F}_{q,i}, \mathbf{u}_n/\tau)\right)\right)}, \quad n = 1, 2, \cdots, S \tag{7}$$

where $\texttt{sim}(\cdot, \cdot)$ is the cosine similarity between two vectors, $\tau$ is a temperate coefficient. The $\max$ operation is critical because it facilitates comparing a feature to its closest background proxy.

Table 1: Comparison with state-of-the-art methods on PASCAL-$5^i$. We report 1-shot and 5-shot results using the mean IoU (%).

| Backbone | Method | 1-shot | | | | | 5-shot | | | | |
|---|---|---|---|---|---|---|---|---|---|---|---|
| | | S0 | S1 | S2 | S3 | Mean | S0 | S1 | S2 | S3 | Mean |
| Res-50 | RPMM [56] | 55.2 | 66.9 | 52.6 | 50.7 | 56.3 | 56.3 | 67.3 | 54.5 | 51.0 | 57.3 |
| | PFENet [46] | 61.7 | 69.5 | 55.4 | 56.3 | 60.8 | 63.1 | 70.7 | 55.8 | 57.9 | 61.9 |
| | CyCTR [64] | 67.8 | 72.8 | 58.0 | 58.0 | 64.2 | 71.1 | 73.2 | 60.5 | 57.5 | 65.6 |
| | HSNet [37] | 64.3 | 70.7 | 60.3 | 60.5 | 64.0 | 70.3 | 73.2 | 67.4 | 67.1 | 69.5 |
| | BAM [26] | 69.0 | 73.6 | 67.6 | 61.1 | 67.8 | 70.6 | 75.1 | 70.8 | 67.2 | 70.9 |
| Res-101 | DAN [49] | 54.7 | 68.6 | 57.8 | 51.6 | 58.2 | 57.9 | 69.0 | 60.1 | 54.9 | 60.5 |
| | RePRI [2] | 59.6 | 68.6 | 62.2 | 47.2 | 59.4 | 66.2 | 71.4 | 67.0 | 57.7 | 65.6 |
| | PFENet [46] | 60.5 | 69.4 | 54.4 | 55.9 | 60.1 | 62.8 | 70.4 | 54.9 | 57.6 | 61.4 |
| | CyCTR [64] | 69.3 | 72.7 | 56.5 | 58.6 | 64.3 | 73.5 | 74.0 | 58.6 | 60.2 | 66.6 |
| | HSNet [37] | 67.3 | 72.3 | 62.0 | 63.1 | 66.2 | 71.8 | 74.4 | 67.0 | 68.3 | 70.4 |
| ViT-B/16 | Baseline | 62.9 | 69.1 | 62.2 | 53.0 | 61.8 | 70.5 | 76.0 | 74.2 | 65.5 | 71.5 |
| | **FPTrans** | **67.1** | **69.8** | **65.6** | **56.4** | **64.7** | **73.5** | **75.7** | **77.4** | **68.3** | **73.7** |
| DeiT-B/16 | Baseline | 68.2 | 69.4 | 61.7 | 60.5 | 64.9 | 75.3 | 78.1 | 76.1 | 73.7 | 75.8 |
| | **FPTrans** | **72.3** | **70.6** | **68.3** | **64.1** | **68.8** | **76.7** | **79.0** | **81.0** | **75.1** | **78.0** |

Given the predicted probability and the ground-truth label, FPTrans uses the standard cross-entropy loss $\mathcal{L}_{ce}$ for supervising the pixel-wise classification. In parallel to $\mathcal{L}_{ce}$, FPTrans uses the prompt-based proxies in a similar procedure and derives another classification loss $\mathcal{L}'_{ce}$.

**Pairwise loss.** In addition to the classification losses, FPTrans further employs a pairwise loss to pull close all the foreground features from the query and support samples, as well as to push the foreground and background features far away from each other. The pairwise loss is formulated as:

$$\mathcal{L}_{pair} = \frac{1}{Z} \sum_{(Y_{q,i}+Y_{s,j}) \geq 1} \text{BCE}(\sigma(\texttt{sim}(\mathbf{F}_{q,i}, \mathbf{F}_{s,j})/\tau), \mathbf{1}[Y_{q,i} = Y_{s,j}]), \tag{8}$$

where $Z = |(Y_{q,i} + Y_{s,j}) \geq 1|$ is the normalization factor and BCE is the binary cross-entropy loss. $\sigma$ is a Sigmoid layer and $\mathbf{1}[\cdot]$ is the indicator function. If two features both belong to the background ($(Y_{q,i} + Y_{s,j}) = 0$), this pairwise loss function will NOT pull them close. Ablation studies show that pulling two foreground features close substantially improves FSS (Table 4) while pulling two background features actually compromises the FSS accuracy, as evidenced in the supplementary Section E.2.

**Overall,** FPTrans sums all the three losses for training:

$$\mathcal{L} = \mathcal{L}_{ce} + \mathcal{L}'_{ce} + \lambda\mathcal{L}_{pair}, \tag{9}$$

where $\lambda$ is a hyperparameter. For inference, FPTrans simply uses the query feature and the feature-based proxies for the final prediction (Eqn. (7))

## 4 Experiments

### 4.1 Implementation Details

**Datasets and Metrics.** We use two popular FSS benchmarks PASCAL-$5^i$ [41] and COCO-$20^i$ [46] for evaluation. PASCAL-$5^i$ combines PASCAL VOC 2012 [10] and SBD [15], and includes 20 classes. Following prior works [46, 26], we split the dataset into four splits with each split 15 classes for training and 5 classes for testing. COCO-$20^i$ is constructed with COCO 2014 [30] and includes 80 classes. It is divided into 4 splits with each split 60 classes for training and 20 classes for testing. To compare with previous methods, we report mean IoU (mIoU) averaged on test classes [46, 64, 26].

**Training Details.** All the images are resized and cropped to $480 \times 480$ and augmented following [46]. We evaluate the proposed method on two vision transformer backbones, ViT-B/16 [9] and DeiT-B/16 [47]. These two backbones are both pretrained on Imagenet-1k [6]. The cross-entropy

Table 2: Comparison with state-of-the-art methods on COCO-$20^i$. We report 1-shot and 5-shot results using the mean IoU (%).

| Backbone | Method | 1-shot | | | | | 5-shot | | | | |
|---|---|---|---|---|---|---|---|---|---|---|---|
| | | S0 | S1 | S2 | S3 | Mean | S0 | S1 | S2 | S3 | Mean |
| Res-50 | RePRI [2] | 32.0 | 38.7 | 32.7 | 33.1 | 34.1 | 39.3 | 45.4 | 39.7 | 41.8 | 41.6 |
| | HSNet [37] | 36.3 | 43.1 | 38.7 | 38.7 | 39.2 | 43.3 | 51.3 | 48.2 | 45.0 | 46.9 |
| | BAM [51] | 43.4 | 50.6 | 47.5 | 43.4 | 46.2 | 49.3 | 54.2 | 51.6 | 49.6 | 51.2 |
| Res-101 | DAN [49] | - | - | - | - | 24.4 | - | - | - | - | 29.6 |
| | PFENet [46] | 34.3 | 33.0 | 32.3 | 30.1 | 32.4 | 38.5 | 38.6 | 38.2 | 34.3 | 37.4 |
| | HSNet [37] | 37.2 | 44.1 | 42.4 | 41.3 | 41.2 | 45.9 | 53.0 | 51.8 | 47.1 | 49.5 |
| ViT-B/16 | Baseline | 37.3 | 39.6 | 41.5 | 35.3 | 38.4 | 48.2 | 53.5 | 52.9 | 48.8 | 50.8 |
| | **FPTrans** | **39.7** | **44.1** | **44.4** | **39.7** | **42.0** | **49.9** | **56.5** | **55.4** | **53.2** | **53.8** |
| DeiT-B/16 | Baseline | 41.8 | 45.4 | 48.8 | 40.3 | 44.1 | 53.9 | 60.1 | 58.9 | 54.4 | 56.8 |
| | **FPTrans** | **44.4** | **48.9** | **50.6** | **44.0** | **47.0** | **54.2** | **62.5** | **61.3** | **57.6** | **58.9** |

Table 3: Evaluation (Mean IoU (%)) under the domain shift from COCO-$20^i$ to PASCAL-$5^i$.

| Method | Backbone | COCO→PASCAL | |
|---|---|---|---|
| | | 1-shot | 5-shot |
| PFENet [46] | Res-50 | 61.1 | 63.4 |
| RePRI [2] | | 63.2 | 67.7 |
| HSNet [37] | Res-101 | 64.1 | 70.3 |
| FPTrans | ViT-B/16 | 67.6 | 76.9 |
| | DeiT-B/16 | **69.7** | **79.3** |

Table 4: Ablation studies. "Pair Loss", "Prompts" and "Proxies" control using (or not using) the pairwise loss, the prompts, and the multiple local background proxies, respectively.

| Pair Loss | Prompts | Proxies | PASCAL | COCO |
|---|---|---|---|---|
| | | | 61.8 | 38.4 |
| ✓ | | | 62.9 | 38.8 |
| ✓ | ✓ | | 63.9 | 41.5 |
| ✓ | | ✓ | 64.0 | 40.3 |
| ✓ | ✓ | ✓ | **64.7** | **42.0** |

losses are optimized with boundary-enhanced weight maps introduced by [65]. We use the SGD optimizer with a momentum of 0.9, a weight decay of 5e-5, and a constant learning rate of 1e-3. Using 4 A100 GPUs, we train 60 epochs with ViT and 30 epochs with DeiT backbone, using a batch size 4 for PASCAL-$5^i$ and 16 for COCO-$20^i$ (batch size 8 in 5-shot due to the memory limitation). When we generate the local background prompts (and the feature-based proxies), the background of each support image is partitioned into 5 local parts, $i.e.$, $S = 5$. Each prompt consists of 12 tokens, $i.e.$, $G = 12$. The weight factor $\lambda$ for balancing the classification loss and pairwise loss (Eqn. (9)) is set as 2e-2 for PASCAL-$5^i$ and 1e-4 for COCO-$20^i$. Our baseline is implemented as the plain vision transformer.

## 4.2 Comparison with State-of-the-Art Methods

**Main results.** Table 1 evaluates the proposed FPTrans on PASCAL-$5^i$, from which we draw two observations as follows:

First, FPTrans consistently improves the baseline on two backbones. For example, when using the ViT-B/16 as its baseline, FPTrans surpasses the baseline by +2.9% and +2.2% mIoU on 1-shot and 5-shot settings, respectively. We note that compared with the baseline, FPTrans has three major differences, $i.e.$, query-support interactions to better utilize the limited support samples, multiple local background proxies to promote novel-class generalization, and an additional pairwise loss function to pull close foreground features. Ablation studies in Table 4 confirm these advantages as the main reasons that improve the baseline.

Second, comparing FPTrans against state-of-the-art methods, we find that FPTrans achieves competitive FSS accuracy. Under the 1-shot setting, FPTrans on DeiT-B/16 surpasses the most competitive BAM [26] by 1.0% mIoU. Under the 5-shot setting, the superiority of FPTrans is even larger, $i.e.$, +2.8% based on ViT-B/16 and +7.1% based on DeiT-B/16.

Table 5: Performance (Mean IoU (%)) of previous methods with transformer backbones. Experiments are conducted on PASCAL-$5^i$.

| Method | Res-50 | Res-101 | ViT | DeiT |
|---|---|---|---|---|
| PFENet [46] | 60.8 | 60.1 | 58.7 | 57.7 |
| CyCTR [64] | 64.2 | 64.3 | 60.1 | 61.0 |
| Baseline | - | - | 61.8 | 64.9 |
| FPTrans | - | - | **64.7** | **68.8** |

Table 6: Results (Mean IoU (%)) of the different number of background proxies $S$ with the proposed methods (Ours) or mixture model (Mix.) [56].

| $S$ | | 1 | 3 | 5 | 7 | 9 |
|---|---|---|---|---|---|---|
| PASCAL-$5^i$ | Ours | 63.9 | 64.4 | **64.7** | 64.4 | 64.0 |
| | Mix. | - | 64.2 | 64.1 | - | - |
| COCO-$20^i$ | Ours | 41.5 | 41.9 | **42.0** | 41.7 | 41.2 |
| | Mix. | - | 40.9 | 40.6 | - | - |

Table 2 summarizes the results on COCO-$20^i$. The major observations are consistent as on PASCAL-$5^i$. FPTrans on ViT-B/16 and DeiT-B/16 both surpass the prior state of the art by a clear margin, setting a new state of the art.

**Domain shift scenario.** Few-shot learning has been studied under a domain shift scenario [22]. Therefore, we also evaluate the proposed FPTrans on the domain shift scenario for semantic segmentation, where the base training data and the testing data have a significant domain gap. We use COCO-$20^i$ for training and use PASCAL-$5^i$ for testing, following previous works [2, 37] for comparison. The training classes (in COCO-$20^i$) and the novel testing classes (in PASCAL-$5^i$) are not overlapped. We summarize the average results on 4 COCO-trained models in Table 3. FPTrans outperforms HSNet [37] by +5.6% and +9.0% on the 1-shot and 5-shot settings, respectively. It confirms the effectiveness of FPTrans under the domain shift scenario.

### 4.3 Ablation Study

To better understand the proposed methods, we conduct ablation studies on FPTrans components. Experiments are conducted with a ViT-B/16 backbone on the 1-shot setting unless specified otherwise.

**Ablations on some major components.** We recall that FPTrans has multiple keypoints / important designs, *i.e.*, an additional pairwise loss for training, a novel prompting strategy, and multiple local proxies. The corresponding ablations are shown in Table 4, from which we draw three observations. First, the pairwise loss improves the baseline by +1.1% gains on PASCAL-$5^i$ and +0.4% on COCO-$20^i$. Second, adding prompts further brings +1.0% and +2.7% gains while adding multiple background proxies further brings +1.1% and +1.5% gains on PASCAL-$5^i$ and COCO-$20^i$, respectively. Third, compared with the baseline, the full FPTrans equipped with all the three components achieves overall improvements of +2.9% and 3.6% on PASCAL-$5^i$ and COCO-$20^i$, respectively.

**Transformer backbone for decoder-based method.** Based on two competitive decoder-based methods PFENet [46] and CyCTR [64], we replace their CNN backbones with the ViT-B/16 and DeiT-B/16 backbones, as shown in Table 5. We observe that these two methods undergo considerable accuracy decreases after the backbone replacement. It suggests that replacing the CNN backbone with a transformer does not necessarily improve FSS, although the transformer backbone fits our plain FSS framework.

**Investigation of the local background proxies.** We investigate the local background proxies by varying their numbers $S$ in Table 6. It is observed that using 5 background proxies achieves the highest accuracy on both PASCAL-$5^i$ and COCO-$20^i$ for our method. Moreover, we compare our method with another multi-proxy method ("Mix.") proposed in RPMM [56]. RPMM online trains mixture models to generate multiple background proxies. In contrast, our approach for achieving multiple background is relatively simple (by partitioning

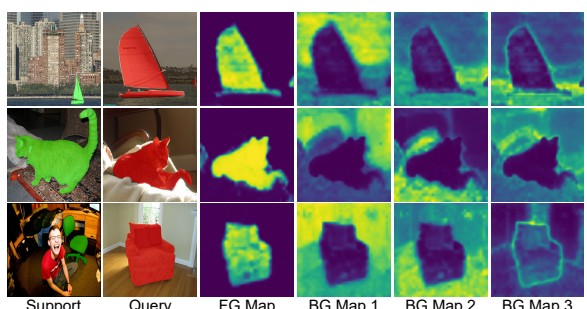

Figure 3: Visualization of predicted score maps. Each background proxy focus on the partial background.

the background) and superior (*e.g.*, +0.6% gains on PASCAL-5$^i$ and +1.4% gains on COCO-20$^i$ when $S = 5$. We infer that our partition-based background proxies are more representative because it considers a realistic factor, *i.e.*, the far-away regions of the background are likely to be inhomogeneous to each other. Fig. 3 visualizes the activation maps of different proxies (one foreground proxy and three local background proxies). It confirms that the local background proxies well accommodate the inhomogeneous background.

## 5 Conclusion

This paper revives the plain framework "feature extractor + linear classification head" and correspondingly proposes a novel Feature-Proxy Transformer (FPTrans) for few-shot segmentation. During feature extraction, FPTrans makes the query interact with support features in all the transformer blocks, therefore well utilizing the limited support samples. During proxy extraction, FPTrans encodes the complex background into multiple local background proxies, therefore improving the generalization towards novel classes. Given the discriminative features and the representative proxies, FPTrans directly uses a linear classification head to compare their cosine similarity and achieves state-of-the-art performance. With its simplicity and competitive accuracy, we hope FPTrans can serve as a strong baseline for few-shot segmentation.

**Limitations and future works.** Currently, FPTrans relies on a pretrained model for generating prompts, which consumes extra computation resources. However, we find that FPTrans is robust to the model for extracting the prompts (see supplementary Section F for details) to some extent. Therefore, we will seek more lightweight prompt-generating models in future works.

## Acknowledgements

The work of the authors was supported by the National Natural Science Foundation of China (No. 62132017), and the Fundamental Research Funds for the Central Universities (No. 226-2022-00087).

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
