# A  A Detailed Revisit to Vision Transformer

Vision transformer (ViT) [5] derived from Transformer [16] is designed for computer vision tasks. It consists of an embedding module, a sequence of stacked transformer blocks, and an MLP head. Specifically, given an RGB image $I$, ViT first crops and reshapes it into a series of image patches $\{\mathbf{a}_p \in \mathbb{R}^{3 \times P \times P}\}_{p=1}^N$ ($P$ is the patch size), and then projects them into $C$-dimensional embeddings:

$$\mathbf{x}_p = \texttt{Embed}(\mathbf{a}_p), \qquad p = 1, 2, \dots, N. \tag{1}$$

We denote the collection of embeddings as $\mathbf{X} = \{\mathbf{x}_p\}_{p=1}^N$.

In each transformer block $B_l$, the layer inputs are processed by a Multiheaded Self-Attention (MSA) module and a multilayer perceptron (MLP) module with extra residual connections. More concretely, given a sequence of input tokens $\mathbf{X} \in \mathbb{R}^{N \times C}$, a self-attention layer corresponds to three matrices: queries $\mathbf{Q} = \mathbf{X}\mathbf{W}^Q$, keys $\mathbf{K} = \mathbf{X}\mathbf{W}^K$ and values $\mathbf{V} = \mathbf{X}\mathbf{W}^V$, where $\mathbf{W}^Q, \mathbf{W}^K \in \mathbb{R}^{C \times d_k}$, and $\mathbf{W}^V \in \mathbb{R}^{C \times d_v}$ are projection weights. With $\mathbf{Q}, \mathbf{K}$ and $\mathbf{V}$, self-attention can be formulated as follows:

$$\text{Attention}(\mathbf{Q}, \mathbf{K}, \mathbf{V}) = \text{Softmax}(\frac{\mathbf{Q}\mathbf{K}^\top}{\sqrt{d_k}})\mathbf{V}. \tag{2}$$

MSA is constructed based on Attention by split the channels of $\mathbf{Q}, \mathbf{K}$ and $\mathbf{V}$ into $h$ groups with each group a part of queries, keys, and values $\mathbf{Q}_i, \mathbf{K}_i \in \mathbb{R}^{N \times \frac{d_k}{h}}, \mathbf{V}_i \in \mathbb{R}^{N \times \frac{d_v}{h}}$. Therefore, MSA concatenates the attentions results of each group by:

$$\text{MSA}(\mathbf{Q}, \mathbf{K}, \mathbf{V}) = [\text{head}_1, \text{head}_2, \cdots, \text{head}_h]\mathbf{W}^O, \tag{3}$$

where $\text{head}_i = \text{Attention}(\mathbf{Q}_i, \mathbf{K}_i, \mathbf{V}_i)$, and $[\cdot, \cdot]$ is the concatenate operation. $\mathbf{W}^O \in \mathbb{R}^{d_v \times C}$ is the projection weights for layer outputs.

With the MSA module and MLP module (a two-layer MLP), the ViT can be formulated as:

$$\mathbf{X}_0 = [\mathbf{x}_{cls}, \mathbf{X}_{img}] + \mathbf{E}_{pos}, \tag{4}$$

$$\mathbf{X}_l' = \text{MSA}(\text{LN}(\mathbf{X}_{l-1})) + \mathbf{X}_{l-1}, \qquad l = 1, 2, \dots, L, \tag{5}$$

$$\mathbf{X}_l = \text{MLP}(\text{LN}(\mathbf{X}_l')) + \mathbf{X}_l', \qquad l = 1, 2, \dots, L, \tag{6}$$

$$\mathbf{y} = \text{MLP\_Head}(\text{LN}(\mathbf{X}_L^0)), \tag{7}$$

where $\mathbf{E}_{pos} \in \mathbb{R}^{(N+1) \times C}$ is the position embedding and LN is the layer normalization [2]. MLP_Head is a linear classifier and $\mathbf{X}_L^0$ is the deep state of the class token $\mathbf{x}_{cls}$. We note that Eqn. (5) and Eqn. (6) comprise the transformer block $B_l$.

# B  Details for Prompt Generation

As illustrated in Section 3.3.2 in the manuscript, we generate a set of prompts from each support image, including a foreground prompt and multiple local background prompts. Generating these local background prompts requires partitioning the background into several local parts. Moreover, we use an augmentation method and derive multiple prompt tokens from each generated prompt. The augmented prompt tokens are then concatenated with the patch tokens and are fed into the transformer blocks. Here we provide some important details for the prompt generation procedure in the proposed FPTrans, *i.e.*, the Voronoi-Based Background partition, and the prompt augmentation.

## B.1  Voronoi-Based Background Partition

FPTrans adopts a Voronoi-based method [1] to partition the background into multiple regions. The partition method consists of three steps:

**Step 1:** We collect all the background positions into a position set $\mathcal{B} = \{t = (i, j)|Y_{s,t} = 0, \ i = 1, 2, \dots, W, \ j = 1, 2, \dots, H\}$, where $Y_s$ is the support mask, and $H$ and $W$ are the height and width, respectively. An empty point list $\mathcal{T}$ is initialized.

**Step 2:** We select $S$ dispersed points from the position set $\mathcal{B}$. Specifically, a seed $t_1$ is first randomly sampled from $\mathcal{B}$ and appended into $\mathcal{T}$. Then, we select from $\mathcal{B}$ the next seed point $t_2$ satisfying:

$$t_2 = \operatorname*{arg\,max}_{t \in \mathcal{B}} \min_{t' \in \mathcal{T}} \|t - t'\|_2^2, \tag{8}$$

which is the farthest point to all the points in $\mathcal{T}$. $t_2$ is appended into $\mathcal{T}$. Repeating Eqn. (8), we can finally select $S$ points dispersed in the background.

**Step 3:** Given these $S$ dispersed points $\mathcal{T} = \{t_1, t_2, \ldots, t_S\}$ as the seeds, we assign the neighboring pixels of each seed into the same part according to the Voronoi diagram and correspondingly derive $S$ local parts. Concretely, for each background pixel $t \in \mathcal{B}$, we assign a label $m_t$ as follows:

$$m_t = \underset{n \in \{1,2,\ldots,S\}}{\arg\min} \|t - t_n\|_2^2. \tag{9}$$

Therefore, the background is divided into $S$ regions formulated as $\mathcal{B}_n = \{t \in \mathcal{B} | m_t = n\}$, $n = 1, 2, \ldots, S$. The point sets $\mathcal{B}_n$ are further transferred into binary masks as $B_n = \{0, 1\}^{H \times W}$ for the subsequent masked average pooling as stated in the Section 3.3.2 of the main manuscript.

### B.2   Prompt Augmentation

With the downsampled support foreground mask $\tilde{Y}_s$ and background partitions $B_n$, $n = 1, 2, \ldots, S$, we calculate the foreground and multiple background mean features by masked average pooling:

$$\mathbf{u}_f^* = \frac{1}{|\tilde{Y}_s|} \sum_{i=1}^{H \times W} \mathbf{F}_{s,i}^* \tilde{Y}_{s,i}, \quad \mathbf{u}_n^* = \frac{1}{|B_n|} \sum_{i=1}^{H \times W} \mathbf{F}_{s,i}^* B_{s,i}, \quad n = 1, 2, \ldots, S, \tag{10}$$

where $\mathbf{F}_s^*$ is the support features extracted by a pretrained ViT. Inspired by the multiple-object tracking within a single framework [21], in which different objects are represented by various identifications (*i.e.*, learnable vectors) for simultaneously tracking, we add extra learnable tokens to the mean features for more discriminative prompts. Specifically, we first expand $C$-dimensional mean features ($\mathbf{u}_f$ and $\mathbf{u}_n$) into a corresponding token $\mathcal{E}(\mathbf{u}_f), \mathcal{E}(\mathbf{u}_n) \in \mathbb{R}^{G \times C}$ (by repeating each mean feature $G$ times). Then, a group of learnable tokens is added to obtain the final prompts:

$$\mathbf{p}_f = \mathcal{E}(\mathbf{u}_f^*) + \mathbf{z}_f, \quad \mathbf{p}_n = \mathcal{E}(\mathbf{u}_n^*) + \mathbf{z}_n, \quad n = 1, 2, \ldots, S. \tag{11}$$

We note that in different episodes, $\mathbf{u}_f^*$ and $\mathbf{u}_n^*$ represent diverse foreground and background classes, which implies that $\mathbf{z}_f$ and $\mathbf{z}_n$ should not be bound to specific classes. To this end, we initialize a learnable token pool $\mathcal{W} = \{\mathbf{z} | \mathbf{z} \in \mathbb{R}^{G \times C}\}$ with size $|\mathcal{W}| = D$. In each episode, $S + 1$ learnable tokens are randomly sampled $\{\mathbf{z}_f, \mathbf{z}_1, \mathbf{z}_2, \ldots, \mathbf{z}_S\} \subset \mathcal{W}$ and used for constructing prompts using Eqn. (11). In this way, these tokens are optimized (by gradient back-propagation from foreground and background prompts) to be diverse from each other, which in turn enhances the discrimination of prompts.

## C   Details for $K$-Shot Setting

FPTrans can be naturally extended to the $K$-shot setting when $K > 1$.

Specifically, for the **prompt generation** (the process of **feature-based proxy extraction** is similar), we calculate a foreground mean feature and $S$ background mean features for each support sample by:

$$\mathbf{u}_f^{(k)} = \frac{1}{|\tilde{Y}_s^{(k)}|} \sum_{i=1}^{HW} \mathbf{F}_{s,i}^{(k)} \tilde{Y}_{s,i}^{(k)}, \quad k = 1, \ldots, K \tag{12}$$

$$\mathbf{u}_n^{(k)} = \frac{1}{|B_n^{(k)}|} \sum_{i=1}^{HW} \mathbf{F}_{s,i}^{(k)} B_{n,i}^{(k)}, \quad k = 1, \ldots, K, \ n = 1, \ldots, S, \tag{13}$$

where $\mathbf{F}_s^{(k)}$ is the support feature, $\tilde{Y}_s^{(k)} \in \{0, 1\}^{H \times W}$ is the down-sampled foreground mask, and $\{B_n^{(k)}\}_{n=1}^S$ are background partitions. The final foreground mean feature (or proxy) is calculated by taking the average on $\{\mathbf{u}_f^{(k)}\}_{k=1}^K$ following prior methods [14, 23]:

$$\mathbf{u}_f = \frac{1}{K} \sum_{k=1}^{K} \mathbf{u}_f^{(k)}. \tag{14}$$

The $K \times S$ background mean features (or proxies) are kept because the backgrounds among support images are also likely to be inhomogeneous, which gives $\mathbf{u}_n$, $n = 1, 2, \ldots, KS$.

For the **feature extraction**, $K$ support samples are individually processed and the prompt synchronization is applied to the $K + 1$ prompts, which is formulated by:

$$[\mathbf{X}_q^l, \mathbf{P}_q^l] = B_l([\mathbf{X}_q^{l-1}, \mathbf{P}^{l-1}]), \tag{15}$$

$$[\mathbf{X}_s^{l,(k)}, \mathbf{P}_s^{l,(k)}] = B_l([\mathbf{X}_s^{l-1,(k)}, \mathbf{P}^{l-1}]), \quad k = 1, 2, \ldots, K \tag{16}$$

$$\mathbf{P}^l = \frac{1}{K+1}\left(\mathbf{P}_q^l + \sum_{k=1}^{K}\mathbf{P}_s^{l,(k)}\right), \tag{17}$$

where $l = 1, 2, \cdots, L$ enumerates all the transformer blocks.

## D  Detailed Experimeantal Settings

### D.1  Datasets

**PASCAL-5$^i$**   is built from PASCAL VOC 2012 [6] (See the website for details.) and SBD [7](See the website for details.). We make the dataset splits following [14], as shown in Table 1.

Table 1: Detailed splits of PASCAL-5$^i$

| Split | Test classes |
|---|---|
| PASCAL-5$^0$ | aeroplane, bicycle, bird, boat, bottle |
| PASCAL-5$^1$ | bus, car, cat, chair, cow |
| PASCAL-5$^2$ | diningtable, dog, horse, motorbike, person |
| PASCAL-5$^3$ | potted plant, sheep, sofa, train, tv/monitor |

**COCO-20$^i$**   is built from COCO 2014 [9] (Licenses of all the images are contained in the annotation file. See the website for details.). We make the dataset splits following [14], as shown in Table 2.

Table 2: Detailed splits of COCO-20$^i$

| Split | Test classes |
|---|---|
| COCO-20$^0$ | Person, Airplane, Boat, Park meter, Dog, Elephant, Backpack, Suitcase, Sports ball, Skateboard, W. glass, Spoon, Sandwich, Hot dog, Chair, D. table, Mouse, Microwave, Fridge, Scissors |
| COCO-20$^1$ | Bicycle, Bus, T.light, Bench, Horse, Bear, Umbrella, Frisbee, Kite, Surfboard, Cup, Bowl, Orange, Pizza, Couch, Toilet, Remote, Oven, Book, Teddy |
| COCO-20$^2$ | Car, Train, Fire H., Bird, Sheep, Zebra, Handbag, Skis, B. bat, T. racket, Fork, Banana, Broccoli, Donut, P. plant, TV, Keyboard, Toaster, Clock, Hairdrier |
| COCO-20$^3$ | Motorcycle, Truck, Stop, Cat, Cow, Giraffe, Tie, Snowboard, B. glove, Bottle, Knife, Apple, Carrot, Cake, Bed, Laptop, Cellphone, Sink, Vase, Toothbrush |

**COCO-20$^i$ $\rightarrow$ PASCAL-5$^i$**   For the domain shift setting, we make the dataset splits following [3], as shown in Table 3.

The construction method of PASCAL-5$^i$ and COCO-20$^i$ follows previous work [14]. We do not find personally identifiable information or offensive content in the two datasets.

### D.2  Implementation Details

We implement FPTrans on vision transformer backbones (*i.e.*, ViT [5], DeiT [15]) and a proxy-based classification head. Our experiments are based on ViT-B/16 and DeiT-B/16 (both are pretrained

Table 3: Detailed splits of COCO-$20^i$ → PASCAL-$5^i$

| Split | PASCAL-$5^i$ Test classes |
|---|---|
| COCO-$20^0$ | Airplane, Boat, Chair, D. table, Dog, Person |
| COCO-$20^1$ | Horse, Sofa, Bicycle, Bus |
| COCO-$20^2$ | Bird, Car, P.plant, Sheep, Train, TV |
| COCO-$20^3$ | Bottle, Cat, Cow, Motorcycle |

on ImageNet-1K [4] with size 224 and finetuned with size 384). We also report the results of two smaller variants, DeiT-S/16 and DeiT-T/16 (pretrained with size 224) in Fig. 1. To enlarge the size of feature maps, we append a residual upsampling layer after the backbone in the baseline and FPTrans. Specifically, given the output of the final transformer block, we reshape it back to 2D feature maps $\mathbf{X}_L \in \mathbb{R}^{C \times H \times W}$, and upsample the feature maps by:

$$\mathbf{X}'_L = \text{Resize}(\mathbf{X}_L) + g(\mathbf{X}_L),\tag{18}$$

where Resize is a linear interpolation operation, and $g(\cdot)$ is a bottleneck layer implemented by "Conv-ReLU-DeConv-ReLU-Conv", where "Conv" is the $1 \times 1$ convolutional layer and "DeConv" is the $2 \times 2$ deconvolutional layer. The hidden channels are set as 256 by default. We observe that a strong pairwise loss (*i.e.*, a large $\lambda$) leads to over-penalization. Therefore, we experimentally set $\lambda$ to 2e-2 for PASCAL-$5^i$ and smaller 1e-4 for a larger dataset COCO-$20^i$.

### D.3 Algorithm

The pseudo-code of FPTrans is presented as below. The PyTorch [13] and PaddlePaddle[1] implementation will be publicly available.

## E More Experimental Results

### E.1 Detailed Main Results

See Table 7 and Table 8.

### E.2 Ablation Studies

**Ablations on the transformer blocks.** Previous methods [18, 22, 14] use mid-level features because high-level features are prone to lack of details. We inspect different feature levels on various transformer backbones as shown in Fig. 1. We observe that FPTrans prefers mid-level features to top-level features, which is consistent with that of CNN-based methods [22, 14]. For example, ViT-B/16 achieves the best results with 10 transformer blocks, while the three DeiT variants (DeiT-B/16, DeiT-S/16, and DeiT-T/16) achieve the best results with 11 blocks. Moreover, The smaller DeiT-S/16 backbone even outperforms the ViT-B/16 backbone, and the smallest DeiT-T/16 backbone with 11 blocks achieves 59.70% mIoU on par with some ResNet-101 methods, *e.g.*, RePRI [3] (59.4%) and PFENet (60.1%). For COCO-$20^i$, we find that DeiT-B/16 with 12 transformer blocks gives the best performance.

**Ablations on prompt augmentation.** The ablation studies of prompts are listed in Table 4. From the results in the table, we have two observations.

- #2 (using only the learnable prompts) and #3 (using only the extracted prompts) actually decreases and increases the accuracy over #1 (baseline), respectively.
- Comparing #4 against #3, we observe that adding the learnable prompts brings further improvement.

Therefore, in our manuscript, we consider the learnable prompts as a prompt augmentation approach (which should not be used alone).

---

[1] https://github.com/PaddlePaddle/Paddle

**Algorithm 1** Algorithm of FPTrans (Single training step, 1-shot setting)

**Require:** A training episodes $(I_s, Y_s, I_q, Y_q)$
1: $\{\mathbf{a}_{q,p}\}_{p=1}^N, \{\mathbf{a}_{s,p}\}_{p=1}^N \leftarrow I_q, I_s$ ▷ Reshape images into image patches
2: $\mathbf{X}_q \leftarrow \texttt{Embed}(\{\mathbf{a}_{q,p}\}_{p=1}^N), \mathbf{X}_s \leftarrow \texttt{Embed}(\{\mathbf{a}_{s,p}\}_{p=1}^N)$ ▷ Project image patches into embedding
3: $[\mathbf{x}_{cls}, \mathbf{X}_q] \leftarrow \mathbf{X}_q, [\mathbf{x}_{cls}, \mathbf{X}_s] \leftarrow \mathbf{X}_s$ ▷ Concatenate the class token
4: $[\mathbf{x}^0, \mathbf{X}_q^0] \leftarrow [\mathbf{x}_{cls}, \mathbf{X}_q] + \mathbf{E}_{pos}, [\mathbf{x}^0, \mathbf{X}_s^0] \leftarrow [\mathbf{x}_{cls}, \mathbf{X}_s] + \mathbf{E}_{pos}$ ▷ Add the positional embedding
5: $\mathbf{P}^0 \leftarrow \textsc{PromptGeneration}([\mathbf{x}_{cls}, \mathbf{X}_s])$ ▷ Prompt generation
6: $l = 1$
7: **while** $l \leq L$ **do**
8: $\quad [\mathbf{x}_q^l, \mathbf{X}_q^l, \mathbf{P}_q^l] \leftarrow B_l([\mathbf{x}^{l-1}, \mathbf{X}_q^{l-1}, \mathbf{P}^{l-1}])$
9: $\quad [\mathbf{x}_s^l, \mathbf{X}_s^l, \mathbf{P}_s^l] \leftarrow B_l([\mathbf{x}^{l-1}, \mathbf{X}_s^{l-1}, \mathbf{P}^{l-1}])$
10: $\quad \mathbf{x}^l \leftarrow (\mathbf{x}_q^l + \mathbf{x}_s^l)/2$
11: $\quad \mathbf{P}^l \leftarrow (\mathbf{P}_q^l + \mathbf{P}_s^l)/2$
12: $\quad l \leftarrow l + 1$
13: **end while**
14: $\mathbf{F}_q \leftarrow \texttt{Resize}(\mathbf{X}_q^L) + g(\mathbf{X}_q^L), \mathbf{F}_s \leftarrow \texttt{Resize}(\mathbf{X}_s^L) + g(\mathbf{X}_s^L)$ ▷ Upsample features
15: $[\mathbf{p}_f^L, \{\mathbf{p}_n^L\}_{n=1}^S] \leftarrow \mathbf{P}^L + g(\mathbf{P}^L)$ ▷ Project prompt states into the feature space
16: $\mathbf{u} := [\mathbf{u}_f, \{\mathbf{u}_n\}_{n=1}^S] \leftarrow \textsc{ProxyGeneration}(\mathbf{F}_s, Y_s)$ ▷ Feature-based proxy generation
17: $\mathbf{v} := [\mathbf{v}_f, \{\mathbf{v}_n\}_{n=1}^S] \leftarrow [\frac{1}{G}\sum_{j=1}^G \mathbf{p}_{f,j}^L, \{\frac{1}{G}\sum_{j=1}^G \mathbf{p}_{n,j}^L\}_{n=1}^S]$ ▷ Prompt-based proxy generation
18: $\mathcal{P}(\mathbf{F}_{q,i}, \mathbf{u}) \leftarrow \frac{\exp(\texttt{sim}(\mathbf{F}_{q,i}, \mathbf{u}_f)/\tau)}{\exp(\texttt{sim}(\mathbf{F}_{q,i}, \mathbf{u}_f)/\tau) + \max_n(\exp(\texttt{sim}(\mathbf{F}_{q,i}, \mathbf{u}_n/\tau)))}$ ▷ Compute probability
19: $\mathcal{L}_{ce} \leftarrow -\sum_{i=1}^{H \times W}(Y_{q,i} \log \mathcal{P}(\mathbf{F}_{q,i}, \mathbf{u}) + (1 - Y_{q,i}) \log(1 - \mathcal{P}(\mathbf{F}_{q,i}, \mathbf{u})))$
▷ Feature-proxy based classification loss
20: $\mathcal{L}_{ce}' \leftarrow -\sum_{i=1}^{H \times W}(Y_{q,i} \log \mathcal{P}(\mathbf{F}_{q,i}, \mathbf{v}) + (1 - Y_{q,i}) \log(1 - \mathcal{P}(\mathbf{F}_{q,i}, \mathbf{v})))$
▷ Prompt-proxy based Classification loss
21: $\mathcal{L}_{pair} = \frac{1}{Z}\sum_{(Y_{q,i} + Y_{s,j}) \geq 1} \text{BCE}(\sigma(\texttt{sim}(\mathbf{F}_{q,i}, \mathbf{F}_{s,j})/\tau), \mathbf{1}[Y_{q,i} = Y_{s,j}]),$ ▷ Pairwise loss
22: $\mathcal{L} \leftarrow \mathcal{L}_{ce} + \mathcal{L}_{ce}' + \lambda \mathcal{L}_{pair}$

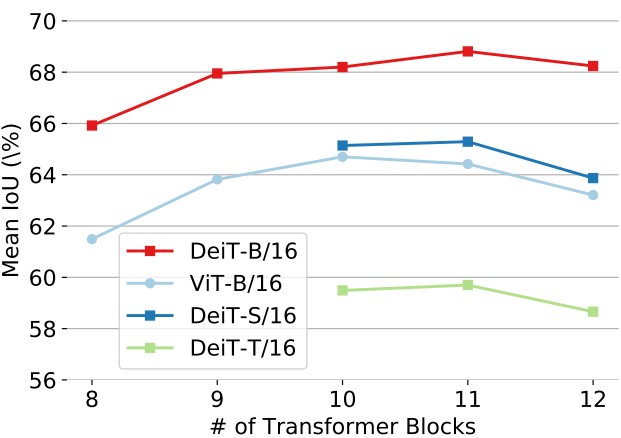

Figure 1: Results with different levels of feature and various backbones on the 1-shot setting. Experiments are conducted on PASCAL-$5^i$.

Table 4: Ablation studies of the prompting strategy. "Extracted prompts" indicate the foreground and background mean vectors extracted in the prompt generation step. "Learned prompts" indicate the extra added learnable vectors. We conduct experiments with ViT-B/16 on the 1-shot setting.

| # | Extracted prompts | Learnable prompts | PASCAL-$5^i$ | COCO-$20^i$ |
|---|---|---|---|---|
| 1 | | | 64.0 | 40.3 |
| 2 | | ✓ | 63.5 | 39.2 |
| 3 | ✓ | | 64.3 | 41.4 |
| 4 | ✓ | ✓ | **64.7** | **42.0** |

**Ablations on the pairwise loss.** We recall that the pairwise loss for training FPTrans does NOT pull close background features. The motivation for this design is that we consider the background is not homogeneous. We conduct experiments to show that enforcing within-class compactness on background features actually compromises FPTrans, as shown in Table 5. It is observed that adding the pulling-close to the background features decreases the segmentation accuracy (-0.7% IoU with 50% background pairs, and -0.9% mIoU with all the background pairs).

Table 5: Results of using the background-background pairs in the pairwise loss. We conduct experiments with ViT-B/16 on the 1-shot setting, and report results on PASCAL-$5^i$.

| Using background-background pairs (percent) | Mean IoU |
|---|---|
| 100% | 63.8 |
| 50% | 64.0 |
| 0% | **64.7** |

### E.3 Qualitative Results

As presented in Fig. 2, we display some qualitative comparisons of FPTrans with the baseline and two previous methods PFENet [14] and BAM [8].

## F   Analysis on Computational Cost

Although FPTrans achieves promising results with a simple framework, it relies on a pretrained model for generating prompts, which consumes extra computational resources. However, the computational cost for generating the prompts is comparable compared with previous methods.

We add the comparison of the parameter size and FLOPs in Table 6. For a fair comparison, we fix the input size as $480 \times 480$. We observe that the proposed **FPTrans is actually relatively efficient**, considering its superiority in FSS accuracy. For example, FPTrans with the DeiT-S/16 backbone has 41 Mb parameters and only 80.7 GFLOPs. It is faster than all the competing CNN methods and yet achieves competitive accuracy. Moreover, FPTrans with DeiT-B/16 backbone is superior to the SOTA method BAM w.r.t. both the accuracy (mIoU) and speed (FLOPs).

Moreover, the computational cost for generating the prompts has the potential to be reduced, because we find that FPTrans is robust to the model for extracting the prompts to some extent. Specifically, when replacing the pretrained ViT with the trained backbone of FPTrans (on PASCAL-$5^i$), the generated prompts can still produce almost the same results. In this way, we only need to save a copy of FPTrans, instead of saving both the pretrained ViT and FPTrans for inference. In future works, we will seek more lightweight prompt-generating models to further reduce the computational cost. Since FPTrans only consists of a transformer backbone and a linear classification head, the quantitative computational cost of FPTrans can be directly referred to as vision transformer [5, 15].

Table 6: Comparison of FPTrans with SOTA methods on the number of parameters and computation cost. * denotes that the mean IoU is from our experiments.

| Backbone | Method | Params (M) | GFLOPs | Mean-IoU (%) |
|---|---|---|---|---|
| ResNet-50 | PFENet [14] | 34 | 231.2 | 60.8 |
| | CyCTR [23] | 37 | 244.7 | 64.2 |
| | HSNet [11] | 28 | 95.9 | 64.0 |
| | BAM [8] | 52 | 302.2 | 67.8 |
| ResNet-101 | PFENet [14] | 53 | 367.9 | 60.1 |
| | CyCTR [23] | 59 | 381.1 | 64.3 |
| | HSNet [11] | 47 | 145.0 | 66.2 |
| | BAM* | 71 | 438.9 | 67.5 |
| ViT-B/16 | | 145 | 247.2 | 64.7 |
| DeiT-T/16 | FPTrans | 11 | 26.7 | 59.7 |
| DeiT-S/16 | | 41 | 80.7 | 65.3 |
| DeiT-B/16 | | 159 | 271.8 | 68.8 |

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

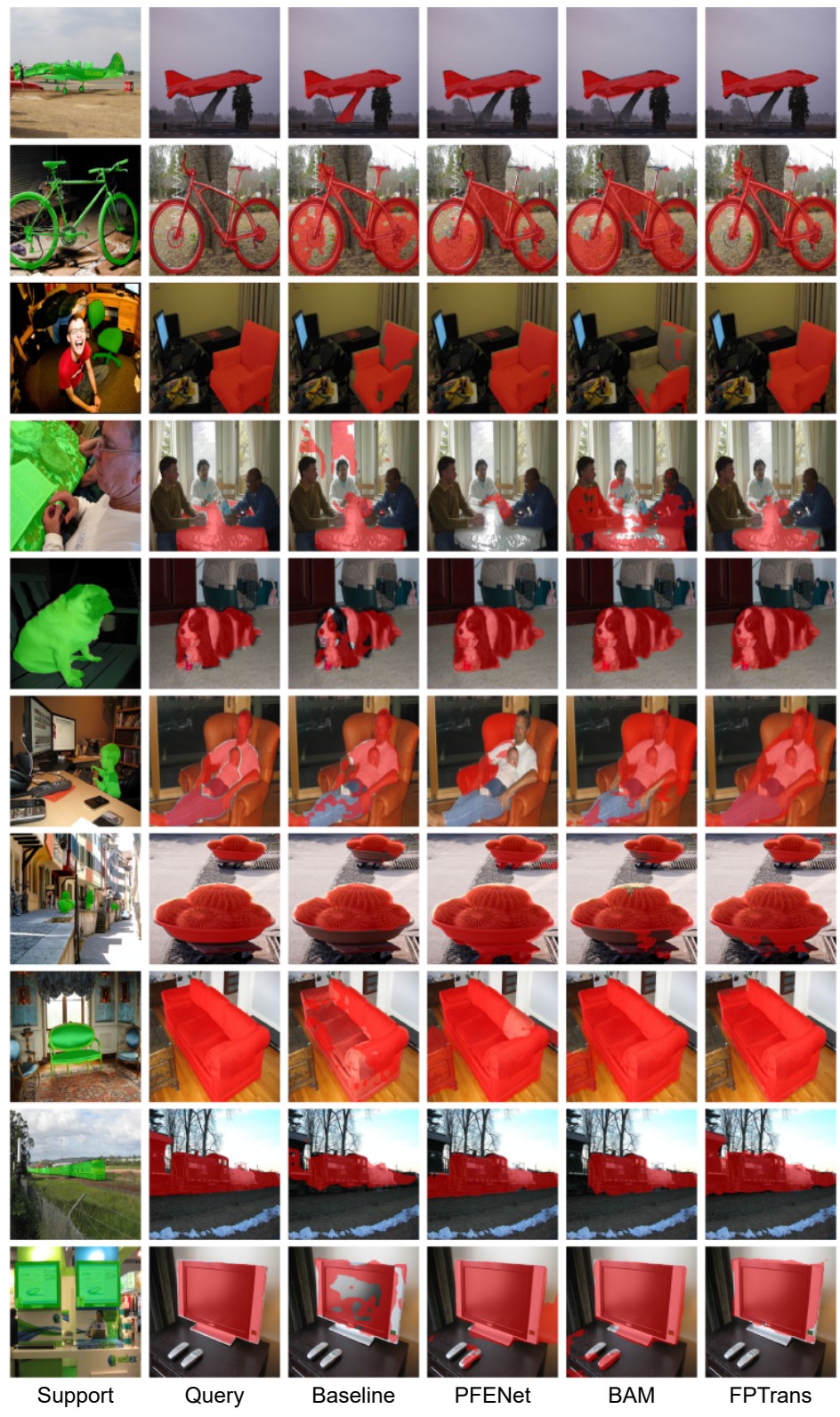

| Support | Query | Baseline | PFENet | BAM | FPTrans |

Figure 2: Qualitative comparisons of FPTrans with the baseline and previous methods, PFENet [14] and BAM [8].

Table 7: Comparison with state-of-the-art methods on PASCAL-$5^i$. We report the mean IoU (%) and standard deviation averaged on three random seeds under 1-shot and 5-shot settings.

| Backbone | Method | 1-shot | | | | | 5-shot | | | | |
|---|---|---|---|---|---|---|---|---|---|---|---|
| | | S0 | S1 | S2 | S3 | Mean | S0 | S1 | S2 | S3 | Mean |
| Res-50 | RPMM [20] | 55.2 | 66.9 | 52.6 | 50.7 | 56.3 | 56.3 | 67.3 | 54.5 | 51.0 | 57.3 |
| | PPNet [10] | 47.8 | 58.8 | 53.8 | 45.6 | 51.5 | 58.4 | 67.8 | 64.9 | 56.7 | 62.0 |
| | PFENet [14] | 61.7 | 69.5 | 55.4 | 56.3 | 60.8 | 63.1 | 70.7 | 55.8 | 57.9 | 61.9 |
| | CyCTR [23] | 67.8 | 72.8 | 58.0 | 58.0 | 64.2 | 71.1 | 73.2 | 60.5 | 57.5 | 65.6 |
| | HSNet [11] | 64.3 | 70.7 | 60.3 | 60.5 | 64.0 | 70.3 | 73.2 | 67.4 | 67.1 | 69.5 |
| | BAM [8] | 69.0 | 73.6 | 67.6 | 61.1 | 67.8 | 70.6 | 75.1 | 70.8 | 67.2 | 70.9 |
| Res-101 | FWB [12] | 51.3 | 64.5 | 56.7 | 52.2 | 56.2 | 54.9 | 67.4 | 62.2 | 55.3 | 59.9 |
| | DAN [17] | 54.7 | 68.6 | 57.8 | 51.6 | 58.2 | 57.9 | 69.0 | 60.1 | 54.9 | 60.5 |
| | RePRI [3] | 59.6 | 68.6 | 62.2 | 47.2 | 59.4 | 66.2 | 71.4 | 67.0 | 57.7 | 65.6 |
| | PFENet [14] | 60.5 | 69.4 | 54.4 | 55.9 | 60.1 | 62.8 | 70.4 | 54.9 | 57.6 | 61.4 |
| | CyCTR [23] | 69.3 | 72.7 | 56.5 | 58.6 | 64.3 | 73.5 | 74.0 | 58.6 | 60.2 | 66.6 |
| | HSNet [11] | 67.3 | 72.3 | 62.0 | 63.1 | 66.2 | 71.8 | 74.4 | 67.0 | 68.3 | 70.4 |
| ViT-B/16 | Baseline | 62.9±0.36 | 69.1±0.06 | 62.2±0.55 | 53.0±0.30 | 61.8±0.07 | 70.5±1.09 | 76.0±0.32 | 74.2±0.46 | 65.5±1.24 | 71.5±0.77 |
| | **FPTrans** | **67.1**±0.47 | **69.8**±0.03 | **65.6**±0.14 | **56.4**±0.75 | **64.7**±0.29 | **73.5**±0.12 | 75.7±0.04 | **77.4**±0.15 | **68.3**±0.19 | **73.7**±0.08 |
| DeiT-B/16 | Baseline | 68.2±0.34 | 69.4±0.38 | 61.7±1.06 | 60.5±0.66 | 64.9±0.11 | 75.3±0.19 | 78.1±0.12 | 76.1±0.55 | 73.7±0.76 | 75.8±0.11 |
| | **FPTrans** | **72.3**±0.49 | **70.6**±0.34 | **68.3**±0.47 | **64.1**±0.49 | **68.8**±0.33 | **76.7**±0.36 | **79.0**±0.03 | **81.0**±0.89 | **75.1**±0.31 | **78.0**±0.14 |

Table 8: Comparison with state-of-the-art methods on COCO-20$^i$. We report the mean IoU (%) and standard deviation averaged on three random seeds under 1-shot and 5-shot settings.

| Backbone | Method | 1-shot | | | | | 5-shot | | | | |
|---|---|---|---|---|---|---|---|---|---|---|---|
| | | S0 | S1 | S2 | S3 | Mean | S0 | S1 | S2 | S3 | Mean |
| Res-50 | RePRI [3] | 32.0 | 38.7 | 32.7 | 33.1 | 34.1 | 39.3 | 45.4 | 39.7 | 41.8 | 41.6 |
| | HSNet [11] | 36.3 | 43.1 | 38.7 | 38.7 | 39.2 | 43.3 | 51.3 | 48.2 | 45.0 | 46.9 |
| | BAM [19] | 43.4 | 50.6 | 47.5 | 43.4 | 46.2 | 49.3 | 54.2 | 51.6 | 49.6 | 51.2 |
| Res-101 | DAN [17] | - | - | - | - | 24.4 | - | - | - | - | 29.6 |
| | PFENet [14] | 34.3 | 33.0 | 32.3 | 30.1 | 32.4 | 38.5 | 38.6 | 38.2 | 34.3 | 37.4 |
| | HSNet [11] | 37.2 | 44.1 | 42.4 | 41.3 | 41.2 | 45.9 | 53.0 | 51.8 | 47.1 | 49.5 |
| ViT-B/16 | Baseline | 37.3±0.31 | 39.6±0.20 | 41.5±0.30 | 35.3±0.35 | 38.4±0.15 | 48.2±0.36 | 53.5±0.16 | 52.9±0.16 | 48.8±0.15 | 50.8±0.20 |
| | **FPTrans** | **39.7**±0.24 | **44.1**±0.17 | **44.4**±0.23 | **39.7**±0.30 | **42.0**±0.06 | **49.9**±0.29 | **56.5**±0.05 | **55.4**±0.08 | **53.2**±0.34 | **53.8**±0.13 |
| DeiT-B/16 | Baseline | 41.8±0.55 | 45.4±0.75 | 48.8±0.19 | 40.3±0.58 | 44.1±0.47 | 53.9±0.53 | 60.1±0.75 | 58.9±0.55 | 54.4±0.80 | 56.8±0.64 |
| | **FPTrans** | **44.4**±0.27 | **48.9**±0.37 | **50.6**±0.16 | **44.0**±0.36 | **47.0**±0.11 | **54.2**±0.36 | **62.5**±0.20 | **61.3**±0.20 | **57.6**±0.27 | **58.9**±0.06 |

[12] Khoi Nguyen and Sinisa Todorovic. Feature weighting and boosting for few-shot segmentation. In *The IEEE International Conference on Computer Vision (ICCV)*, October 2019.

[13] Adam Paszke, Sam Gross, Soumith Chintala, Gregory Chanan, Edward Yang, Zachary DeVito, Zeming Lin, Alban Desmaison, Luca Antiga, and Adam Lerer. Automatic differentiation in PyTorch. In *NIPS-W*, 2017.

[14] Zhuotao Tian, Hengshuang Zhao, Michelle Shu, Zhicheng Yang, Ruiyu Li, and Jiaya Jia. Prior Guided Feature Enrichment Network for Few-Shot Segmentation. *IEEE Trans. Pattern Anal. Mach. Intell.*, pages 1–1, 2020.

[15] Hugo Touvron, Matthieu Cord, Matthijs Douze, Francisco Massa, Alexandre Sablayrolles, and Herve Jegou. Training data-efficient image transformers & distillation through attention. In *Proceedings of the 38th International Conference on Machine Learning*, pages 10347–10357. PMLR, July 2021.

[16] Ashish Vaswani, Noam Shazeer, Niki Parmar, Jakob Uszkoreit, Llion Jones, Aidan N. Gomez, Łukasz Kaiser, and Illia Polosukhin. Attention is all you need. In *NeurIPS*, pages 5998–6008, 2017.

[17] Haochen Wang, Xudong Zhang, Yutao Hu, Yandan Yang, Xianbin Cao, and Xiantong Zhen. Few-Shot Semantic Segmentation with Democratic Attention Networks. In Andrea Vedaldi, Horst Bischof, Thomas Brox, and Jan-Michael Frahm, editors, *ECCV*, Lecture Notes in Computer Science, pages 730–746, Cham, 2020. Springer International Publishing.

[18] Kaixin Wang, Jun Hao Liew, Yingtian Zou, Daquan Zhou, and Jiashi Feng. PANet: Few-shot image semantic segmentation with prototype alignment. In *The IEEE International Conference on Computer Vision (ICCV)*, October 2019.

[19] Zifeng Wang, Zizhao Zhang, Chen-Yu Lee, Han Zhang, Ruoxi Sun, Xiaoqi Ren, Guolong Su, Vincent Perot, Jennifer Dy, and Tomas Pfister. Learning to Prompt for Continual Learning. *CVPR*, March 2022.

[20] Boyu Yang, Chang Liu, Bohao Li, Jianbin Jiao, and Qixiang Ye. Prototype Mixture Models for Few-Shot Semantic Segmentation. In Andrea Vedaldi, Horst Bischof, Thomas Brox, and Jan-Michael Frahm, editors, *ECCV*, volume 12353, pages 763–778. Springer International Publishing, Cham, 2020.

[21] Zongxin Yang, Yunchao Wei, and Yi Yang. Associating Objects with Transformers for Video Object Segmentation. In *NeurIPS*, 2021.

[22] Chi Zhang, Guosheng Lin, Fayao Liu, Rui Yao, and Chunhua Shen. CANet: Class-agnostic segmentation networks with iterative refinement and attentive few-shot learning. In *The IEEE Conference on Computer Vision and Pattern Recognition (CVPR)*, June 2019.

[23] Gengwei Zhang, Guoliang Kang, Yi Yang, and Yunchao Wei. Few-shot segmentation via cycle-consistent transformer. *NeruIPS*, 34, 2021.