# OpenReview forum: "Feature-Proxy Transformer for Few-Shot Segmentation"
_NeurIPS.cc/2022/Conference — NeurIPS 2022 Accept_

### Official Review · Reviewer_Qm9S · 2022-06-27

**Rating:** 6
**Confidence:** 4
**Soundness:** 3 good
**Presentation:** 3 good
**Contribution:** 3 good

**Summary:**

The paper proposes a different approach to FSS, whereby instead of independently processing and matching support and query samples, the support samples are processed to generate prompts, which are then concatenated with both query and support tokens. Importantly these prompts act as message passers between the support and query feature extractors. Finally, proxies are generated and the query tokens classified. The authors claim that this approach has multiple benefits,
1) Architecture is simplified, compared with decoder architectures
2) Performance is on par with or greater than previous approaches

**Questions:**

* Is there a particular reason that you chose not to use a decoder? In my experience, using a simple feature + conditioning based decoder on the predictions can increase segmentation accuracy by quite a large amount. I do not agree that refraining from any upsampling is a simplification of the method, but I'm open to hear your reasoning.


**Limitations:**

The authors addressed the issue of having to use a separate network for the prompt generation.

**Strengths And Weaknesses:**

## Major Strengths
* Novelty, specifically the prompting approach and message passing for conditioning has to my knowledge not previously been done in FSS. I think the approach is interesting and may yield further insights.

## Minor Strengths
* Paper is clearly written and easy to follow.
* Extensive experiments, in particular ablations.

## Major Weaknesses
* It is not easy to judge whether the improvements come from the proposed method, as the backbones have been changed. The authors have provided reimplementations of previous methods using the ViT backbone, however one has to assume that such a change may require finetuning (e.g. learning rates) for those previous methods and hence the comparison is not simple to make. Judging from vision transformers success in other related tasks, such as segmentation and detection, I believe there is a large risk that most (if not all) gains can be attributed to the change of backbone.

## Minor Weaknesses
* While the authors argue that their approach is simpler than previous decoder-based methods, I do not personally agree. To me, the architecture seems quite involved, in particular the need for a first proxy generation step, separate from the rest.
* Code not provided, however, promised to be released.
* Performance gains relative to recent work are not significant, in particular, compared to BAM.

## Conclusions
While there are potential issues with where the performance gains come from, the authors also do not claim this as their main contribution. Instead, it is to be viewed as an alternate way to perform FSS. I think the paper is interesting enough to warrant acceptance, despite my concerns.

---

> ### Author Response · Authors · 2022-08-02
> **Response to Reviewer Qm9S**
>
> Thanks for your valuable comments. We will explain your concerns point by point.
>
> ---
> **Q1**: Is there a reason that you chose not to use a decoder? In my experience, using a simple feature + decoder can largely increase segmentation accuracy. I do not agree that refraining from upsampling is a simplification.
>
> **A1:** As explained in the introduction, our overall motivation for not using the decoder-based framework is: we observe that in the recent SOTA methods, the decoder becomes more and more complicated (much more complicated than just upsampling, as revisited in Sec. 2.1 in the manuscript). Under this background, we think this pipeline is difficult to follow/improve and might soon reach a plateau. We would like to explain this viewpoint in detail as below:
>
> Specifically, while these decoders try to exploit the matching capacity of the already-given features, they barely pay an effort to improve the feature itself. In fact, most decoder-based methods use the fixed pretrained CNN models as the feature extractor. We think directly using the pretrained feature is likely to restrict the upper bound of the FSS accuracy. Interestingly, although transformer are the potential to provide higher discriminative ability, our implementation of "transformer + decoder" does not achieve improvements. Therefore, we revive the plain framework and focus on learning discriminative features and representative classification proxies (in FPTrans, the transformer backbone is fine-tuned rather than being fixed). We think that based on this plain framework, future research might be relatively easier to achieve further improvement. We agree with you that the "transformer + decoder" framework is also potential to bring improvement (maybe with some necessary modification and hyper-parameter optimization). We are open to this alternative strategy and look forward to the corresponding progress from the research community.
>
> ---
> **Q2:** It is not easy to judge whether the improvements come from the proposed method, as the backbones have been changed.
>
> **A2:** We guess your concern is not about our improvement over the transformer baseline, but about the comparison between FPTrans and SOTA CNN-based methods, i.e., why FPTrans has higher performance. We think two reasons are important to the superiority of FPTrans, i.e., strong transformer backbone and our improvement over the transformer baseline. On the one hand, the transformer is indeed a strong backbone for learning discriminative features, e.g., the DeiT baseline achieves 64.9% mIoU on PASCAL, which is already comparable with some SOTA methods. On the other hand, based on the strong transformer baseline, FPTrans further brings considerable improvement (e.g., +3.7% mIoU on PASCAL), consequently setting up a new state of the art.
>
> Since the transformer baseline already maintains relatively high accuracy, achieving further improvement is not easy: we tried our best to adapt existing decoder-based methods to transformer backbones and have not achieved improvements. Moreover, during rebuttal, we find a concurrent work CATrans[r1] shares a similar observation. Specifically, CATrans investigates both ResNet-101 and Swin Transformer as the backbone, and finds that the superiority of using Swin Transformer is only marginal (e.g., +0.6% on PASCAL). Therefore, it is valuable for FPTrans to improve the transformer baseline with a relatively simple framework.
>
> ---
> **Q3:** The architecture seems quite involved, in particular the need for a proxy generation step.
>
> **A3:** We agree that using a prompt generation step increases the complexity over the transformer baseline (as discussed in the limitation part of the manuscript). Given the observation in the supplementary that the prompt generation is robust to some extent, we think this step is the potential to be simplified in future work. Moreover, despite the extra generation step, FPTrans is still quite efficient compared with SOTA methods. For example, FPTrans with DeiT-S/16 achieves competitive results with 80.7 FLOPs, only 1/5 of BAM on ResNet-101, as shown in Table r2 in the response to Reviewer d8mB.
>
> ---
> **Q4:** Code not provided, however, promised to be released.
>
> **A4:** The code and pretrained models will be openly accessible.
>
> ---
> **Q5:** Performance gains relative to recent work are not significant, in particular, compared to BAM.
>
> **A5:** Compared with the strongest competitor BAM, although the performance gain under the 1-shot setting is relatively small (e.g., +1.0% and +0.8% on PASCAL-5i and COCO-20i, respectively), the performance gain becomes much larger when there are more support samples.  For instance, in the 5-shot setting, FPTrans surpasses BAM by +7.1% and +7.7% on PASCAL-5i and COCO-20i, respectively. Moreover, FPTrans has less computation cost as explained in the response of Q3.
>
> ---
> [r1] S. Zhang, et. al, CATrans: Context and Affinity Transformer for Few-Shot Segmentation, 2022, arXiv:2204.12817

---

### Official Review · Reviewer_Ro1k · 2022-07-08

**Rating:** 5
**Confidence:** 4
**Soundness:** 3 good
**Presentation:** 3 good
**Contribution:** 3 good

**Summary:**

This paper proposes the Feature-Proxy Transformer for few-shot segmentation, which utilizes the ViT as the backbone and interacts the query features with support features through prompts. Besides, this paper introduces to obtain multiple background proxies via Voronoi-based method. Experiments are conducted on Pascal-$5^i$ and COCO-$20^i$ datasets.

**Questions:**

In the rebuttal, I would like to see more discussion about the main contribution of this work and the analysis of the prompts. From my perspective, the Pair Loss and Multiple Proxies are minor contributions that are effective but lack novelty. The Prompts for FSS is novel, but the absolute gain (considering the computational cost) is relatively small. It would be better to include the mentioned ablation study, and also the quantitative computational cost comparison (*eg.*, inference time, flops et. al.) in the rebuttal.

**Limitations:**

Limitations are disscussed in the paper.

**Strengths And Weaknesses:**


**Strengths**
1. This paper is easy to follow and may be easy to implement.
2. This paper demonstrates that the framework with ViT backbone could achieve decent performance on both benchmarks.
3. The proposed operations are effective according to the experiments.
4. Interacting query and support features through prompts is interesting.
5. It shows an excellent transferable performance (from COCO to Pascal).

**Weaknesses**
It is hard to tell the main contribution of this work.
First, the idea of extracting multiple proxies (prototypes) from support features is not novel since it has been discussed in [30,52]. The only difference is that this paper applies the Voronoi-based method instead of the EM algorithm [30,52].

Besides, although this paper claims that the FPTrans is a plain framework, the prompt generation process is too heavy. It requires to use the entire ViT backbone to produce the prompts, which largely increases computational cost. Specifically, this additionally adds an approximate 50% computational overhead compared with the baseline, which may be unacceptable since the ViT-Base backbone is already a much larger model compared with Resnet-50/101, but the overall framework can only obtain comparable results with previous approaches with ResNet-50 backbone. Besides, this paper misses the ablation of only using the learnable prompts.

---

> ### Author Response · Authors · 2022-08-02
> **Response to Reviewer Ro1k**
>
> Thanks for your valuable comments. We will explain your concerns point by point.
>
> ---
> **Q1:** Discussion about the main contribution.
>
> **A1:** Our main contributions lie in three aspects.
>
> 1. With a rethink of prior state-of-the-art FSS methods, we find that they all fix the backbone for feature extraction and design more and more complicated decoders (Section 2.1 and Figure 1 in the manuscript). Fixing the backbone restricts the feature discriminative ability and makes it difficult to achieve further improvement by designing more sophisticated decoders. Therefore, we revive the plain "feature extractor + classifier" framework and focus on learning discriminative features and representative proxies.
>
> 2. We integrate two keypoints, i.e., query-support interaction and multiple local background proxies, into the proposed FPTrans. Although using multiple local background proxies is not very novel (we did not claim this as a novel point either), these two keypoints both rely on a novel prompting strategy. All the reviewers positively recognize this prompting strategy and the query-support interaction.
>
> 3. We show that this plain and relatively simple framework achieves competitive FSS accuracy, compared with the decoder-based methods. More specifically,  FPTrans achieves comparable (sometimes even better) performance with relatively fewer FLOPs (see the answer to Q2) compared with SOTA methods. Moreover, FPTrans can better handle the domain shift problem (+5.6% and +9.0% in the 1-shot and 5-shot settings on COCO->PASCAL, compared with HSNet).
>
> ---
> **Q2:**  The prompt generation process is too heavy. The large computational cost makes the absolute gain seems relatively small.
>
> **A2:** The computational cost of the proposed FPTrans is actually relatively small, although it has a prompt generation process. The quantitative analysis is summarized in the second table for R1-Q1. For example, FPTrans with the DeiT-S/16 backbone requires only 80.7 GFLOPs and is faster than all the competing CNN methods (while maintaining competitive accuracy). Moreover, FPTrans with DeiT-B/16 backbone is superior to the SOTA method BAM w.r.t. both the accuracy (mIoU) and speed (FLOPs).
>
> The reason for our relatively low computational cost is that FPTrans uses much smaller feature maps. Specifically, the CNN-based methods use the dilated convolutions (in ResBlock3 and ResBlock4) to get the large $60\times 60$ sized feature maps, which significantly increases the computational cost of the original CNN backbone by about $4\sim 5 \times$. In contrast, FPTrans on the DeiT/B-16 backbone requires a relatively small feature map ( $30\times 30$ sized feature maps) and achieves competitive accuracy with high computational efficiency.
>
> ---
> **Q3:** The ablation of only using the learnable prompts and the analysis of the prompts.
>
> **A3:** The ablation studies of prompts are listed in Table r3. From the results in the table above, we have two observations.
>
> 1. #2 (using only the learnable prompts) and #3 (using only the extracted prompts) actually decreases and increases the accuracy over #1 (baseline), respectively.
>
> 2. Comparing #4 against #3, we observe that adding the learnable prompts brings further improvement.
>
> Therefore, in our manuscript, we consider the learnable prompts as a prompt augmentation approach (which should not be used alone).
>
> Table r3. Ablation studies of the prompting strategy. "Extracted prompts" indicate the foreground and background mean vectors extracted in the prompt generation step. "Learned prompts" indicate the extra added learnable vectors.
>
> | # | Extracted prompts | Learnable prompts | PASCAL-5i | COCO-20i |
> |:---:|:---:|:---:|:---:|:---:|
> | 1 |  |  | 64.0 | 40.3 |
> | 2 |  | ✓ | 63.5 | 39.2 |
> | 3 | ✓ | | 64.3 | 41.4 |
> | 4 | ✓ | ✓ | 64.7 | 42.0 |

---

> > ### Comment · Reviewer_Ro1k · 2022-08-06
> > **Reviewer Response**
> >
> > Thank the authors for their response.
> >
> > I further have one question after reading the rebuttal. As stated by the authors in A1-Point1 and the paper, "previous works typically fixed the pretrained backbone parameters to alleviate overfitting". Therefore, the authors claim that the proposed method can revive the plain "feature extractor + classifier" framework without fixing the backbone network.
> > I am curious why the ViT-based backbone does not encounter the overfitting problem even if it contains much more parameters. For instance, in Tabel 2r, 159M compared with 31~71M ResNet-based methods. In the paper, even the baseline with a plain vision transformer can achieve SOTA results (56.8 vs. 51.1 on COCO 5-shot). The Feature-Proxy and Prompts seem to have nothing to do with this surprising and counterintuitive result.
> >
> > Besides, about Q2, the authors do not directly answer my question. I still think the authors should provide the **relative** computation overhead the proposed prompts strategy brings rather than the overall Flops of FPTrans. Could the authors provide the **inference time and flops** comparison between the model "without Prompting" (Line 1 in Table r3) and "with Prompting" (Line 4 in Table r3)?  It would be better if relative performance gains were also included in this comparison.

---

> > > ### Author Response · Authors · 2022-08-07
> > > **Response to Reviewer Ro1k**
> > >
> > > Thanks for the further comments.
> > >
> > > ---
> > > **Q4:** The CNN backbones are typically fixed to alleviate overfitting. In contrast, the ViT backbone is much larger and yet shows resistance against overfitting. Even the baseline with a plain vision transformer can achieve SOTA results. Why is that?
> > >
> > > **A4:** We did not claim that the transformer backbone does not encounter the overfitting problem. Instead, we believe that the transformer backbone also has an overfitting risk. In the manuscript, we wrote this sentence "previous works typically fixed the pretrained backbone parameters to alleviate overfitting" according to PFENet[42]. Given your concern, we feel this statement might be inaccurate because we are not sure whether the overfitting problem is the only reason for current CNN-based methods to fix the backbone. Moreover, we think it is possible that CNN-based methods will benefit from fine-tuning the backbones with some still-unknown solutions. Therefore, we will remove the statement "to alleviate overfitting".
> > >
> > > As for the phenomenon that the transformer backbone seems stronger than the CNN backbone, It is possible that the transformer does has better generalization ability (in spite of the larger model size) under the few-shot learning task, compared with the CNN model. It is because the model size is not the only factor that determines the generalization ability (or overfitting risk). One evidence to support this argument is: a recent work[r2] observes that the transformers "are strong few-shot learners" (compared with the CNN models).
> > >
> > > ---
> > > **Q5:** The relative computation overhead the proposed prompts strategy brings, and the inference time and flops comparison between the model "without prompting" and "with prompting". It would be better if relative performance gains were also included in this comparison.
> > >
> > > **A5:** Thanks for these good suggestions. In Table r4, we compare using prompt (#2) and no prompt (#1) w.r.t. the relative improvements, parameters, FLOPs, and throughput. To better understand the prompt generation process, we employ a light prompt generator using the first 3 blocks in ViT (#3). This light prompt generator significantly reduces the computation overhead and still achieves similar improvement. In conclusion, the prompt generation process has the following characteristics:
> > > * It does require extra computation costs for representing class-aware information in each episode, as discussed in the limitation part of the manuscript.
> > > * However, we find that using a much smaller prompt generator can maintain similar improvement and effectively reduce the computation overhead.
> > > * Moreover, the prompt generator is used in an off-the-shelf manner. As generating prompts only need support samples, the prompts can be offline precomputed to further accelerate the online inference process.
> > >
> > > Table r4. Ablation studies on the prompt generator. ('ep/sec' indicates episodes per second.)
> > >
> > > | # | Model | Prompt Generator | PASCAL-5i | $\Delta$ | COCO-20i | $\Delta$ | Params(M) | GFLOPs | Throughput (ep/sec) |
> > > | --- | --- | --- | --- | --- | --- | --- | --- | --- | --- |
> > > | 1 | No Prompt | - | 64.0 |  | 40.3 |  | 72.8 | 156.0 | 24 |
> > > | 2 | Large Prompt Generator | 10-block ViT | 64.7 | +0.7 | 42.0 | +1.7 | 145 | 247.2 | 15 |
> > > | 3 | Light Prompt Generator | 3-block ViT | 64.6 | +0.6 | 41.8 | +1.5 | 95.4 | 193.7 | 19 |

---

### Official Review · Reviewer_d8mB · 2022-07-11

**Rating:** 6
**Confidence:** 3
**Soundness:** 3 good
**Presentation:** 3 good
**Contribution:** 2 fair

**Summary:**

Unlike mainstream methods that involve intricate decoder structure, this paper proposes to revive the plain framework of feature extractor + linear classification head for few-shot segmentation (FSS). Concretely, this paper designs a framework called Feature-proxy Transformer (FPTrans), where it utilizes proxy, a vector to represent the foreground as well as the background classes. It makes queries interact with support features and use multiple background proxies through a novel prompting strategy. Extensive experiment results verify that the proposed method achieves competitive performance, which shows that it can serve as a simple yet strong baseline for FSS.

**Questions:**

My questions and suggestions are listed in weaknesses. My main concern lies in the experiment results.

**Strengths And Weaknesses:**

##### Strengths #####
1) This paper is well-organized and easy to read. Figure 2 is very clear to illustrate the whole framework.

2) This paper incorporates prompt learning to few-shot segmentation in a clever way. The framework provides dedicated modifications to make it more suitable for FSS, for instance, multiple background prompts.

3) The visualization results in the paper are good.

##### Weaknesses #####
I have some questions about the experiment results.
1) Network backbone. The comparison is conducted with different network backbones, i.e., the previous methods take ResNet-50 / 101, while the proposed method takes ViT / DeiT. I notice that the author compares PFENet and CyCTR on ViT. However, the experiments on DeiT are not included. And the comparsions on more SOTA methods, e.g., BAM, HSNet are also omitted. In addition, I think it is better to add the parameter size and FLOPS of different backbones for a clearer comparison.
2) 1-shot performance. I notice that the proposed FPTrans shows inferior performance than BAM when taking ViT as the backbone. Can the authors have more discussions or explanations about this phenomenon? In addition, I am curious about the performance of BAM with ResNet-101, which is also an important baseline.

Minor issues:
The text in the figures can be larger to make it clearer.

---

> ### Author Response · Authors · 2022-08-02
> **Response to Reviewer d8mB**
>
> Thanks for your valuable comments. We will explain your concerns point by point.
>
> ---
> **Q1:** Using the DeiT backbone for PFENet, CyCTR, and two SOTA methods BAM and HSNet for better comparison. Comparisons on the parameter size and FLOPs are also recommended.
>
> **A1:** Thanks for these good suggestions.
>
> 1. We first add the comparison against these SOTA methods on PASCAL-5i, using both the ViT and DeiT backbones in Table r1. It is observed that: **i)** The proposed FPTrans surpasses all the compared methods on both the ViT and DeiT backbones. **ii)** These SOTA methods undergo considerable decreases when they change the backbone from CNN to transformer (ViT, DeiT), which is consistent with the observation in the manuscript.
>
> Table r1. Comparison of FPTrans with SOTA methods using ResNet and transformer backbones. The * indicates that the results are not officially reported but are achieved by running their source code on the ResNet-101 backbone.
>
> | Method | ResNet-50 | ResNet-101 | ViT-B/16 | DeiT-B/16 |
> |:---:|:---:|:---:|:---:|:---:|
> | PFENet | 60.8 | 60.1 | 58.7 | 57.7 |
> | CyCTR | 64.2 | 64.3 | 60.1 | 61.0 |
> | HSNet | 64.0 | 66.2 | 53.6 | 61.8 |
> | BAM | 67.8 | 67.5* | 59.3 | 50.1 |
> | Baseline | - | - | 61.8 | 64.9 |
> | FPTrans | - | - | 64.7 | 68.8 |
>
>
> 2. We further add the comparison of the parameter size and FLOPs in Table r2. For a fair comparison, we fix the input size as  $480\times 480$. We observe that the proposed **FPTrans is actually relatively efficient**, considering its superiority in FSS accuracy. For example, FPTrans with the DeiT-S/16 backbone has 41 Mb parameters and only 80.7 GFLOPs. It is faster than all the competing CNN methods and yet achieves competitive accuracy. Moreover, FPTrans with DeiT-B/16 backbone is superior to the SOTA method BAM w.r.t. both the accuracy (mIoU) and speed (FLOPs).
>
> Table r2. Comparison of FPTrans with SOTA methods on the number of parameters and computation cost.
>
> | Backbone | Method | Params (M) | GFLOPs | Mean-IoU (%) |
> |:---:|:---:|:---:|:---:|:---:|
> | ResNet-50 | PFENet | 34 | 231.2 | 60.8 |
> |  | CyCTR | 37 | 244.7 | 64.2 |
> |  | HSNet | 28 | 95.9 | 64.0 |
> |  | BAM | 52 | 302.2 | 67.8 |
> | ResNet-101 | PFENet | 53 | 367.9 | 60.1 |
> |  | CyCTR | 59 | 381.1 | 64.3 |
> |  | HSNet | 47 | 145.0 | 66.2 |
> |  | BAM* | 71 | 438.9 | 67.5 |
> | ViT-B/16 | FPTrans | 145 | 247.2 | 64.7 |
> | DeiT-T/16 | FPTrans | 11 | 26.7 | 59.7 |
> | DeiT-S/16 | FPTrans | 41 | 80.7 | 65.3 |
> | DeiT-B/16 | FPTrans | 159 | 271.8 | 68.8 |
>
>
> ---
> **Q2:** FPTrans with ViT backbone shows inferior 1-shot performance than BAM.
>
> **A2:** We note that the superiority of BAM partially comes from the model ensemble (a base-learner + a meta-learner). Since model ensemble generally brings extra improvement, we think our FPTrans achieving comparable accuracy without model ensemble is also valuable. Moreover, when using DeiT as the backbone, FPTrans maintains higher (68.8%) mIoU and lower computational cost (271.8 GFLOPs), compared with BAM on ResNet-50 (67.8% mIoU and 302.2 GFLOPs) or BAM on ResNet-101 (67.5% mIoU and 438.9 GFLOPs).
>
> ---
> **Q3:** The performance of BAM with ResNet-101.
>
> **A3:** BAM-ResNet101 achieves 67.5% mean IoU on PASCAL-5i, which is slightly lower than BAM-ResNet50 (67.8%). Similar observation (i.e., using the smaller CNN backbone is slightly better) can be observed on PFENet, as well.

---

> > ### Comment · Reviewer_d8mB · 2022-08-08
> > **Raise My Rating**
> >
> > The rebuttal solves my concerns well, so I raise my rating to 6.

---

### Meta-Review · Area_Chair_r8yf · 2022-08-27

**Recommendation:** Accept
**Confidence:** Less certain

**Metareview:**

This paper studies the plain segmentation framework (feature extractor + linear classification) for few-shot segmentation. It introduces a prompt based query and support interaction method to enable this framework to work well. All the reviewers recognize the proposed method is novel and the performance is good. Though they have some concerns on the computational cost and the fairness of the experiment comparison (e.g., whether they use the same backbone), the authors address these concerns well in their response. All the reviewers agree with accepting this submission. Although their ratings are not very strong supportive, AC agrees this submission brings some values to the community. It inspire some new thinking of the FSS framework design. The overall framework is still heavy. Hopefully, in the future follow up works, the framework can be further simplified.

**Award:**

No

---

### Decision · Program_Chairs · 2022-09-14

Accept